# Structure and *in situ* organisation of the *Pyrococcus furiosus* archaellum machinery

Bertram Daum[1,2,3]*, Janet Vonck[1], Annett Bellack[4], Paushali Chaudhury[5], Robert Reichelt[4], Sonja-Verena Albers[5], Reinhard Rachel[4], Werner Kühlbrandt[1]

[1]Max Planck Institute of Biophysics, Frankfurt, Germany; [2]Living Systems Institute, University of Exeter, Exeter, United Kingdom; [3]College of Physics, Engineering and Physical Science, University of Exeter, Exeter, United Kingdom; [4]Institute of Microbiology and Archaea Centre, University of Regensburg, Regensburg, Germany; [5]Institute of Biology II, Molecular Biology of Archaea, University of Freiburg, Freiburg, Germany

**Abstract** The archaellum is the macromolecular machinery that Archaea use for propulsion or surface adhesion, enabling them to proliferate and invade new territories. The molecular composition of the archaellum and of the motor that drives it appears to be entirely distinct from that of the functionally equivalent bacterial flagellum and flagellar motor. Yet, the structure of the archaellum machinery is scarcely known. Using combined modes of electron cryo-microscopy (cryoEM), we have solved the structure of the *Pyrococcus furiosus* archaellum filament at 4.2 Å resolution and visualise the architecture and organisation of its motor complex *in situ*. This allows us to build a structural model combining the archaellum and its motor complex, paving the way to a molecular understanding of archaeal swimming motion.

*For correspondence: b.daum2@exeter.ac.uk

## Introduction

Archaea are ubiquitous microorganisms that thrive in diverse habitats around the globe, ranging from extreme environments, such as boiling hot springs, salt lakes or marine hyperthermal vents to ambient environments, such as fresh water springs (*Perras et al., 2014*) and the human body (*Lurie-Weinberger and Gophna, 2015*). While extremophilic Archaea are of great interest for biotechnology, Archaea that are part of the microbiome of the human digestive system have been implicated in obesity (*Mbakwa et al., 2015*). Fundamental to their prolific diversification is the capability of archaeal cells to move through liquid media. In the course of evolution, Archaea have developed their own locomotion machinery called the archaellum, which is distinct from bacterial and eukaryotic flagella in terms of molecular composition and mode of action (*Albers and Jarrell, 2015*).

Whilst eukaryotic and bacterial flagella have been studied in detail (*Krüger and Engstler, 2015*; *Chen et al., 2011*; *DeRosier, 2006*), less is known about the architecture and assembly of their archaeal counterparts. In the past decade, biochemical investigation and conventional negative-stain electron microscopy have established that the archaellum consists of a helical array of multiple copies of 1–5 archaellins with N-terminal sequence homology to bacterial type 4 pili (T4P)(*Albers and Jarrell, 2015*; *Pohlschroder et al., 2011*). In contrast to bacterial flagella, recent cryoEM data have shown that archaella are not hollow, indicating that they have to be assembled at their base (*Albers and Jarrell, 2015*; *Poweleit et al., 2016*). Furthermore, it has been shown for the Archaeon *Halobacterium salinarum* that swimming motion is driven by ATP hydrolysis instead of ion fluxes (*Streif et al., 2008*) and single-molecule studies have recently demonstrated that archaella rotate in a stepwise manner reminiscent of torque-generating ATP synthases (*Kinosita et al., 2016*).

A multi-protein motor complex located in the cell envelope and powered by ATP hydrolysis drives assembly and rotation of the of the FlaA/FlaB filament (*Albers and Jarrell, 2015*; *Chaudhury et al., 2016*). Biochemistry and bioinformatics suggest that the core of this complex is formed by the AAA motor ATPase FlaI, a putative regulator FlaH, a membrane-bound platform protein FlaJ and the periplasmic FlaG, and FlaF proteins (*Albers and Jarrell, 2015*; *Chaudhury et al., 2016*; *Reindl et al., 2013*). In Crenarchaeota, FlaH is surrounded by a larger ring of FlaX proteins (*Banerjee et al., 2013*), which are absent in Euryarchaeota. Instead, Euryarchaeota encode the additional cytosolic proteins FlaC and FlaD/E (*Albers and Jarrell, 2015*; *Chaudhury et al., 2016*; *Näther-Schindler et al., 2014*), which in *H. salinarum* interact with chemoreceptors via adaptor proteins (*Schlesner et al., 2009*). Whilst the X-ray structure of individual subunits from *Sulfolobus acidocaldarius* FlaF (*Banerjee et al., 2015*), FlaH (*Chaudhury et al., 2016*) and FlaI (*Reindl et al., 2013*) have been solved, the arrangement of the assembled motor complex is unknown.

*Pyrococcus furiosus* is a hyperthermophilic Euryarchaeon that has been isolated from volcanic marine sediments (*Fiala and Stetter, 1986*). Its genome has been fully sequenced (*Robb et al., 2001*) and various tools for genetic manipulation are available (*Kreuzer et al., 2013*; *Waege et al., 2010*; *Lipscomb et al., 2011*). *P. furiosus* is therefore of great interest for basic and applied research and widely used as archaeal model organism. *P. furiosus* is highly motile (*Herzog and Wirth, 2012*), grows at an optimum temperature of 95–100°C (*Fiala and Stetter, 1986*) and assembles numerous archaella on the cell pole (*Fiala and Stetter, 1986*; *Näther et al., 2006*). In *P. furiosus* wild type strains, the fla operon encodes for the three archaellins $FlaB_0$, $FlaB_1$ and $FlaB_2$, of which $FlaB_0$ is by far the most abundant (*Näther-Schindler et al., 2014*). At present it is not clear whether the three proteins can all be part of the same archaellum or if each forms distinct filaments of their own. Electron microscopy of freeze-fracture replica and scanning electron micrographs of cells grown on a variety of solid supports showed that *P. furiosus* archaella establish surface contacts as well as cable-like cell-cell connections, suggesting that they also function in surface attachment, intercellular communication and biofilm formation (*Näther et al., 2006*).

We used electron cryo-tomography, sub-tomogram averaging, single-particle cryoEM and helical reconstruction to obtain an *in situ* map of the *P. furiosus* archaellar motor complex and a near-atomic resolution structure of its archaellum. In combination, our results provide a first model of the architecture of the archaellar locomotion machinery.

## Results

### Electron cryo-tomography of *Pyrococcus furiosus*

Initial cryoEM showed that *P. furiosus* cells grown in rich medium, measure ~1–2 μm across (*Figure 1—figure supplement 1A–B*) and are thus too thick and too dense for electron cryo-tomography. By cultivating *P. furiosus* on minimal SME pyruvate medium, the cell size was reduced to 500–1000 nm (*Figure 1—figure supplement 1C–D*). Tomograms of these cells revealed details of their morphology and interior. In roughly 80% of the cells, a bundle of up to 50 archaella protrudes from a bulge at the cell pole (*Figure 1A,B*). On the EM grid, cells occur individually or in pairs in a range of division states (*Figure 1—figure supplement 2A–D*). Interestingly, cell division of *P. furiosus* initially manifests itself in the formation of two indentations at opposite cell poles (*Figure 1—figure supplement 2B*). Subsequently, one of the two indentations develops into a deep cleft that finally spans the entire cytosol (*Figure 1—figure supplement 2C*). Towards the end of cell division, the two daughter cells are connected by a narrow cytosolic bridge, which is surrounded by both a plasma membrane and an S-layer (*Figure 1—figure supplement 2D*). The presence of archaella in *P. furiosus* seems to depend on the division state. Whilst 86% of non-dividing cells exhibit archaella (*Figure 1—figure supplement 2E*), this is the case for only ~54% of dividing cells, of which one or both show an archaellar bundle (*Figure 1—figure supplement 2F*). Furthermore, most of the dividing cells with archaella are in an early state of division (*Figure 1—figure supplement 2B*).

*P. furiosus* archaellar filaments have a constant diameter of ~11 nm but vary considerably in length (a few 100 nm to several μm). Each filament crosses the S-layer and spans the ~34 nm periplasmic gap (*Figure 1B*). As observed previously (*Briegel et al., 2017*), the archaella traverse the periplasm at variable angles of 60–90° between the filament axis and the membrane plane (*Figure 1—figure supplement 3*). In the tomograms, the filaments emerge from a basal density on the

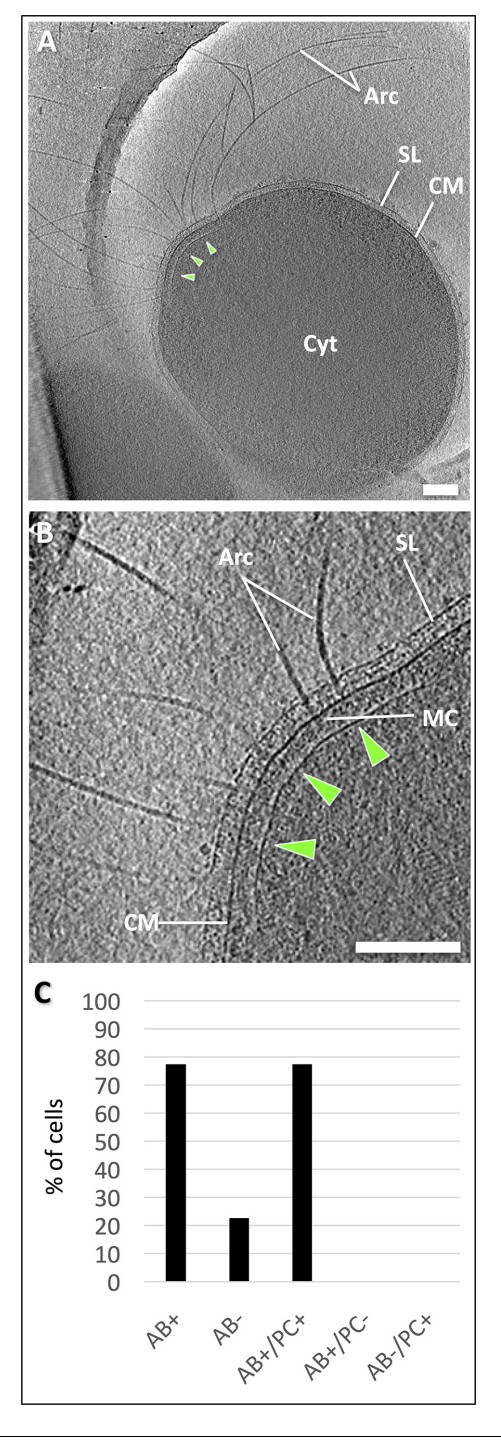

**Figure 1.** Electron cryo-tomography of *P. furiosus*. (**A**) tomographic slice through a frozen-hydrated *P. furiosus* cell. Arc, archaella; SL, S-layer; CM, cell membrane; Cyt, cytosol; green arrowheads, polar cap. (**B**) close-up of the tomogram in A, showing archaella on the cell pole. MC, motor complex. Scale bars, 200 nm. (**C**) percentage of total archaellar bundles observed as well as archaellar bundles observed with and without a polar cap.

*Figure 1 continued on next page*

cytoplasmic surface of the plasma membrane, which most likely corresponds to the archaellar motor (*Figure 1B*).

Each of these densities is located in a ~35 nm gap between the membrane and a sheet-like cytoplasmic structure (polar cap; *Figure 1A,B*). The polar cap appears to be a hallmark of motile Euryarchaeota, as it has also been observed in related species (*Briegel et al., 2017*; *Kupper et al., 1994*). The polar cap has a thickness of ~3 nm, a variable diameter of 200–600 nm and typically straight edges. The edges end in the cytoplasm and are not continuous with the plasma membrane. In its centre, the polar cap is often kinked into a sharp ridge (Figure 4A). We observed the polar cap in all cells in which archaella were present and never in the absence of an archaellar bundle (*Figure 1C*).

## Architecture of the archaellar motor complex

To analyse the *in situ* structure of the archaellar motor complex, we performed sub-tomogram averaging of 379 densities at the base of archaellar filaments. Six-fold symmetry was applied during averaging, as the *S. acidocaldarius* homolog of the central core protein FlaI has been shown to be a hexamer (*Reindl et al., 2013*). To validate this assumption for *P. furiosus*, we overexpressed *Pfu*FlaI in *E. coli*, purified the protein, recorded negative stain images and performed 2D classification of 130.000 particles, which indeed indicated a hexameric ring structure (*Figure 2—figure supplement 1*).

The sub-tomogram average of the motor assembly reveals a bell-shaped central density projecting 19 nm from the plasma membrane into the cytosol (*Figure 2A,B*). Proximal to the membrane, the complex has its narrowest diameter of 9.5 nm, and is linked to the bilayer via six ~3 nm protrusions (*Figure 2A*, arrowheads). Distal to the membrane, the complex widens to a diameter of 18 nm and is connected to a cytosolic ring of 26 nm in cross-section (*Figure 2A–D*).

## Integration of the archaellum into the periplasm

The periplasmic protein FlaF has been suggested to integrate the archaellum filament into the S-layer (*Banerjee et al., 2015*), however it is not certain that this protein forms a *bona fide* conduit through the periplasm. Although the central motor complex is well resolved, both the filament and the regular S-layer structure were averaged

*Figure 1 continued*

The following figure supplements are available for figure 1:

**Figure supplement 1.** CryoEM of *P. furiosus* grown in full medium vs. pyruvate minimal medium.

**Figure supplement 2.** Tomographic slices of *P. furiosus* in different putative division states.

**Figure supplement 3.** Angular freedom of archaella in the periplasm.

out (*Figure 2A*). This indicates that the filament and motor assembly are not structurally aligned with the S-layer, consistent with the variable angle at which the archaellum crosses the periplasm (*Figure 1—figure supplement 3*). Whilst some periplasmic densities proximal to the membrane were resolved (*Figure 2A*), it is not clear if they correspond to an assembly of FlaF or vestiges of the flexible filament. In any case, our data do not show a defined conduit-like-complex that spans the entire periplasm.

Since S-layers are two-dimensional porous protein lattices (*Engelhardt, 2007*), we wondered if the S-layer pores themselves can form conduits for archaellar filaments. To find out if the *P. furiosus* S-layer indeed fulfils this role,

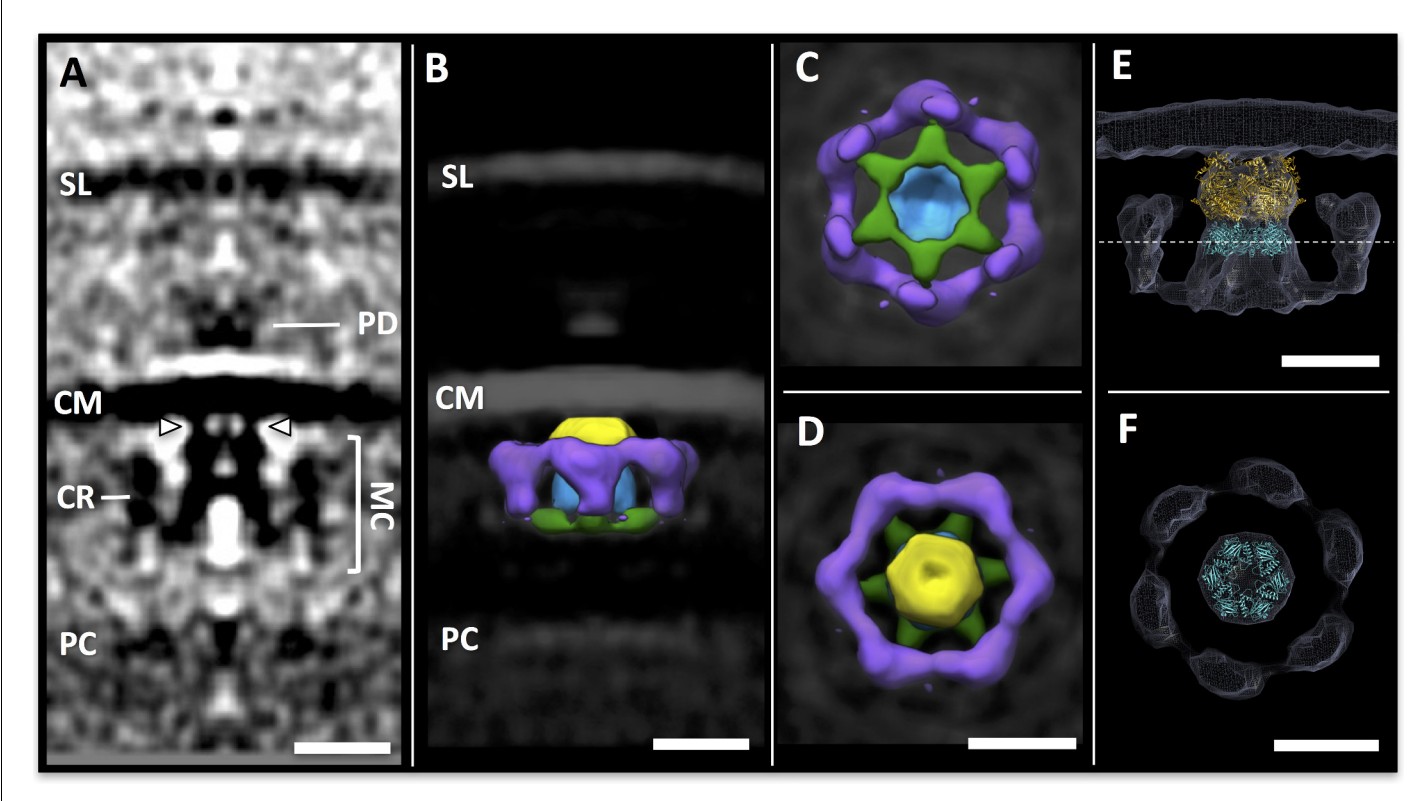

**Figure 2.** Sub-tomogram averaging of the archaellar motor complex. (**A**) tomographic slice through the sub-tomogram average of the motor complex. SL, S-layer; PD; periplasmic densities; CM, cell membrane; MC, motor complex; CR, cytosolic ring; PC, polar cap. Arrowheads indicate two of six narrow connections between MC and CM. (**B–D**) segmented 3D surface representation of the sub-tomogram average of the MC (multiple colours) as seen from the side (**B**), the cytosol (**C**) and the cell membrane (**D**). Yellow, blue, green, central complex; purple, cytosolic ring. (**E,F**) *S. acidocaldarius* FlaI (PDB-4IHQ, gold) and Symmdoc model of *S. acidocaldarius* FlaH (PDB-4YDS, cyan) fitted into the MAC density in side view (**E**) and in cross-section through FlaH (**F**); position of cross-section shown as dotted line in E. Scale bars, 20 nm.

The following figure supplements are available for figure 2:

**Figure supplement 1.** Purification and negative stain EM of *P. furiosus* FlaI.

**Figure supplement 2.** Fourier Shell Correlation (FSC) of MC sub-tomogram average.

1951 S-layer units from *P. furiosus* were aligned and averaged. The resulting map (*Figure 3A,B*; *Figure 3—figure supplement 1*) shows a dense protein array of symmetry P1 and lattice parameters of a = 19 nm and b = 17 nm, including an angle of 63.5°. Note that these parameters are inconsistent with the P6 symmetry that has been deduced from freeze-etched *P. furiosus* cells before (*König et al., 2007*). As our average has been generated from frozen-hydrated cells, we conclude that they represent the native state of the S-layer, whereas the P6 symmetry suggested earlier (*König et al., 2007*) may be a misinterpretation due to limited resolution.

In contrast to the large pores that are often found in S-layers of other archaeal species (*Engelhardt, 2007*; *Veith et al., 2009*), the *P. furiosus* S-layer has narrow, irregular gaps with a maximum width of 3.5 nm. These gaps are evidently too narrow to accommodate the ~11 nm filament, indicating that the S-layer itself does not form a specific guiding scaffold for the archaellum and suggesting that individual S-layer subunits would need to be locally disassembled for the archaellum to pass through. Comparing the S-layer structure to the filament width in silico, we found that removal of a single S-layer subunit would generate sufficient space for the archaellum (*Figure 3C*). Symmetry breaks and gaps are common in S-layers, as they enable them to form closed cages around cells and are essential for cell division (*Pum et al., 1991*). However, the mechanism by which the S-layer lattice is locally dissolved to provide sufficient space for the growing archaellum filament is still unknown and awaits further investigation.

## Location of motor subunits

Previous work has established that the central core of the archaeal motor assembly consists of a complex of the membrane-integral platform protein FlaJ and the soluble, cytosolic ATPase FlaI (*Reindl et al., 2013*). FlaJ is almost completely buried in the lipid bilayer (*Reindl et al., 2013*). Due to the high apparent contrast of lipid bilayers in tomograms at high defocus, it was invisible in our average (*Figure 2A*). However, a superposition of the X-ray structure of the highly conserved *S. acidocaldarius* FlaI (*Sac*FlaI; PDB-4IHQ) with the membrane-proximal cytosolic part of the sub-tomogram average places the X-ray structure into a density that closely matches its outline. Docking *Sac*FlaI into this position also places the six N-terminal domains of FlaI into the six bridging densities between the central complex and the membrane (*Figure 2A,E*). This is in good agreement with the current working model of the motor complex, which suggests that these domains act as binding sites for FlaJ (*Reindl et al., 2013*).

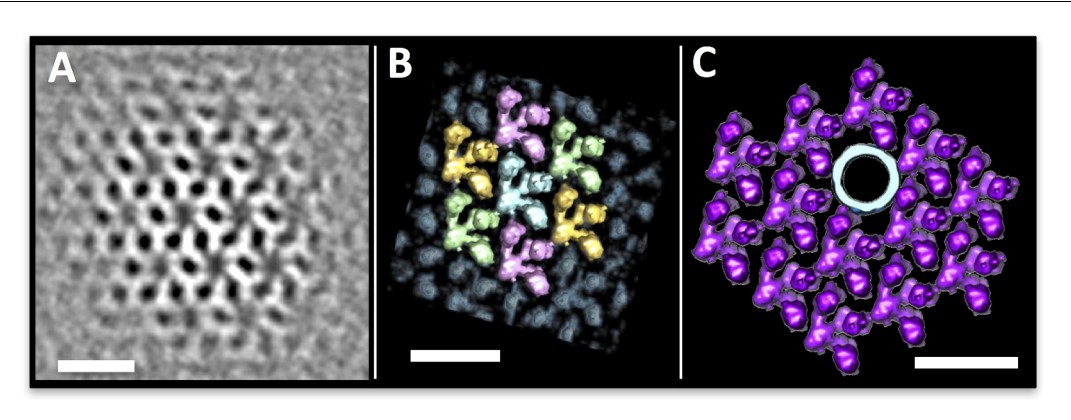

**Figure 3.** The *P. furiosus* S-layer. (**A–C**) sub-tomogram averaging of the *P. furiosus* S-layer as tomographic slice (**A**), segmented 3D surface representation with asymmetric units in different colours (**B**) and one subunit replaced by a 3D surface-rendered sub-tomogram of a *P. furiosus* archaellum (**C**; purple, S-layer; light blue, archaellum). Scale bars, 20 nm.
The following figure supplement is available for figure 3:

**Figure supplement 1.** - Resolution estimate of *P. furiosus* S-layer sub-tomogram average.

According to recently published in vitro binding experiments, FlaI also interacts with FlaH that most likely forms hexameric rings (*Chaudhury et al., 2016*). We thus generated a 6-fold symmetric model of the homologous *S. acidocaldarius* FlaH (*Sac*FlaH; PDB-4YDS) and docked it directly adjacent to *Sac*FlaI (*Figure 2E,F*). This superimposes *Sac*FlaH with a second rounded density that has the dimensions of the hexamer. The resulting model of the FlaI-FlaH complex suggests that FlaH binds as a ring on the cytosolic face of FlaI, in line with previous biochemical data (*Chaudhury et al., 2016*).

After docking FlaI and FlaH into our sub-tomogram averaging map, a large portion of the motor average remains unassigned, especially in its ring region (*Figure 2E,F*). These densities are most likely occupied by the remaining cytosolic subunits FlaC and FlaD/E encoded in the *P. furiosus fla*-operon. Higher resolution structures of these proteins need to be obtained to reliably interpret the density of the cytosolic ring.

## Sub-cellular organisation of motor complexes and the polar cap

By repositioning the motor assembly averages into tomographic volumes, we visualise their positions and mutual arrangement in whole *P. furiosus* cells (*Figure 4A,B*). Strikingly, all motor assemblies co-localise with the polar cap, and they are not observed outside this region (*Figure 4B*).

Close inspection of the polar cap revealed patches of ~9.5 nm hexagonal particles projecting ~15 nm from its intracellular face (*Figure 4C,D*). Sub-tomogram averaging of 57 units shows that these particles have a sandwich-like structure and assemble into hexagonal arrays (*Figure 4C,D*; *Figure 4— figure supplement 1*), with a centre-to-centre distance between closest neighbours of 14 nm. Particles neighbouring the central density are less well resolved (*Figure 4C*), indicating that the organisation of the array is flexible. Although the identity of protein complexes associated with the polar cap is unknown, their localisation in close proximity to the basal bodies suggests that they are mechanistically linked to archaellar function.

## Structure of the *P. furiosus* archaellum

In order to determine the structure of the *P. furiosus* archaellar filament, we collected 297 dose-fractionated movies of isolated fibres and calculated a 3D map using helical reconstruction in Relion 2 (*Kimanius et al., 2016*; *He and Scheres, 2017*). The resulting map has an overall resolution of 4.2 Å (*Figure 5—figure supplement 1*) and reveals a 110 Å-wide helix with a helical rise of 5.41 Å and a rotation of 108.03˚ (*Figure 5A–C*, *Table 1*). The backbone of the archaellum is formed by a bundle of α-helices, surrounded by an outer layer rich in β-strands (*Figure 5B*). The repeating unit of the helical array is a lollipop-shaped protein with an extended inward-facing α-helix and an outward-facing globular domain of several ß-strands (*Figure 5B*), consistent with secondary structure predictions for all three FlaB proteins of the *P. furiosus fla*-operon (*Figure 5—figure supplements 3*).

To investigate which of the three FlaB proteins constitute the *P. furiosus* archaellum, we attempted to build atomic models of $FlaB_0$, $FlaB_1$ and $FlaB_2$, using their respective sequences. Guided by large resolved side chains (*Figure 5D and E*) and the structure of the *M. hungatei* homolog (*Poweleit et al., 2016*) as a reference, we were able to build a model based on the $FlaB_0$ sequence (*Table 1*), but not on $FlaB_1$ or $FlaB_2$.

Whilst our density map fitted $FlaB_0$ well, it could not be reconciled with the inserts found in $FlaB_1$ or $FlaB_2$ (*Figure 5—figure supplements 3*). This suggests that at least the main part of the filament is composed of $FlaB_0$ rather than a combination of all three FlaB proteins, in accordance with previous biochemical data that identified $FlaB_0$ as the major archaellin of *P. furiosus* (*Näther-Schindler et al., 2014*). It remains to be investigated, if the minor archaellins $FlaB_1$ and $FlaB_2$ form specific basal or terminal segments of the $FlaB_0$ filament or assemble into distinct filaments of their own.

Our density map allowed the $FlaB_0$ polypeptide to be modelled from Ala6 onwards, in accordance with posttranslational removal of the short, positively charged N-terminal MAKKG peptide prior to $FlaB_0$ filament assembly. This signal peptide fits the archaeal consensus cleavage site for class III signal peptides [KRDE][GA][ALIFQMVED][ILMVTAS] (*Esquivel et al., 2013*). The structure of the $FlaB_0$ monomer consists of an N-terminal α-helix of 47 amino acids (A6 – A52). The first 25 amino acids of the helix (A6 – I30) are hydrophobic, whereas the second half (Q31 – A52) contains predominantly polar residues and is therefore highly hydrophilic (*Figure 5H*). The TMHMM server predicts

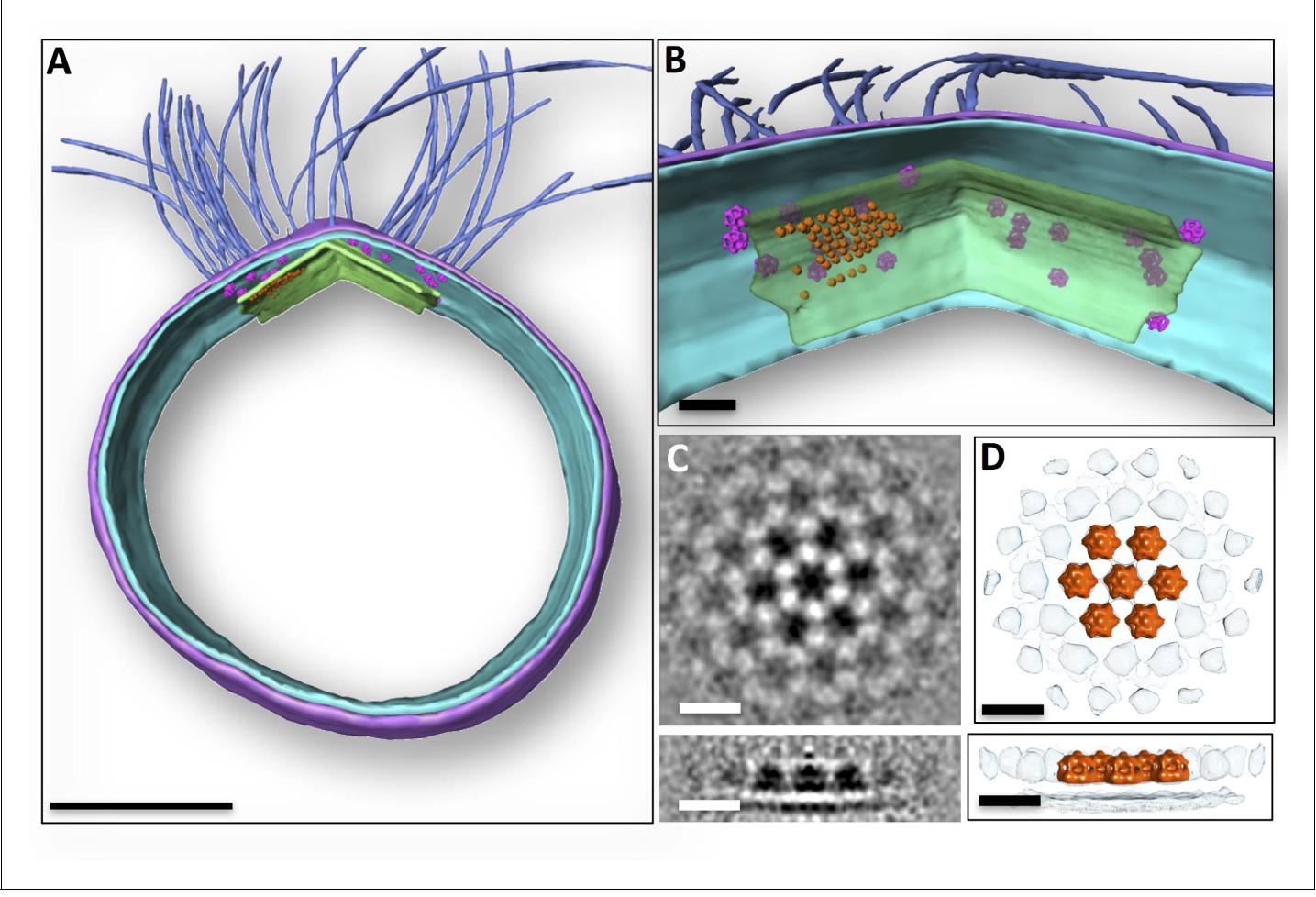

**Figure 4.** Subcellular organisation of motor complexes and polar cap. (**A**) Segmented 3D representation of a tomogram of a *P. furiosus* cell. Motor complexes (magenta) have been repositioned into the original tomogram using coordinates from sub-tomogram averaging. Medium blue, archaella; purple, S-layer; cyan, cell membrane; green, polar cap. (**B**) close-ups of the polar region showing layer-like superimposition of motor complexes, polar cap and hexagonal protein array (orange). Note that due to limitations of manual particle picking only subsets of the motor complexes and hexagonal protein arrays are displayed. (**C** and **D**) sub-tomogram average of hexameric protein array associated with polar cap as slices through the average (**C**), as well as segmented surface representation (**D**) in top (top panel) and side view (bottom panel). Scale bars, 200 nm (**A**); 50 nm (**B**); 15 nm (**C, D**).
The following figure supplement is available for figure 4:

**Figure supplement 1.** Sub-tomogram averaging and resolution of hexagonal protein array.

most of the hydrophobic sections (V7-L29) as trans-membrane, and the subsequent hydrophilic part of the polypeptide as outward-facing (*Figure 5—figure supplement 4*). This suggests that in the monomeric, extra-filamentous form of $FlaB_0$, V7-L29 may act as a transmembrane helix, while the remaining domains protrude into the periplasm. The major part of the polypeptide (S53 - Q212) contains 15 β-strands that fold into a twisted β-barrel (*Figure 6D*) with a hydrophobic interior (*Figure 5D,H*).

Upon integration into the filament, individual $FlaB_0$ subunits interact with one another in multiple ways. The apolar N-terminal α-helix section establishes hydrophobic interactions with six of its neighbours. The resulting α-helix bundle is buried inside the filament and forms its backbone (*Figure 5H, I*). As the α-helices are arranged at an angle of ~15° to the long axis of the archaellum, their hydrophilic C-terminal sections are close to the periphery, where they contact neighbouring α-helices and β-strands via hydrogen bonds and electrostatic interactions (*Figure 5H–I*). In the outer sheath of the archaellum, each β-barrel interacts with six surrounding neighbours, mainly by hydrophilic contacts

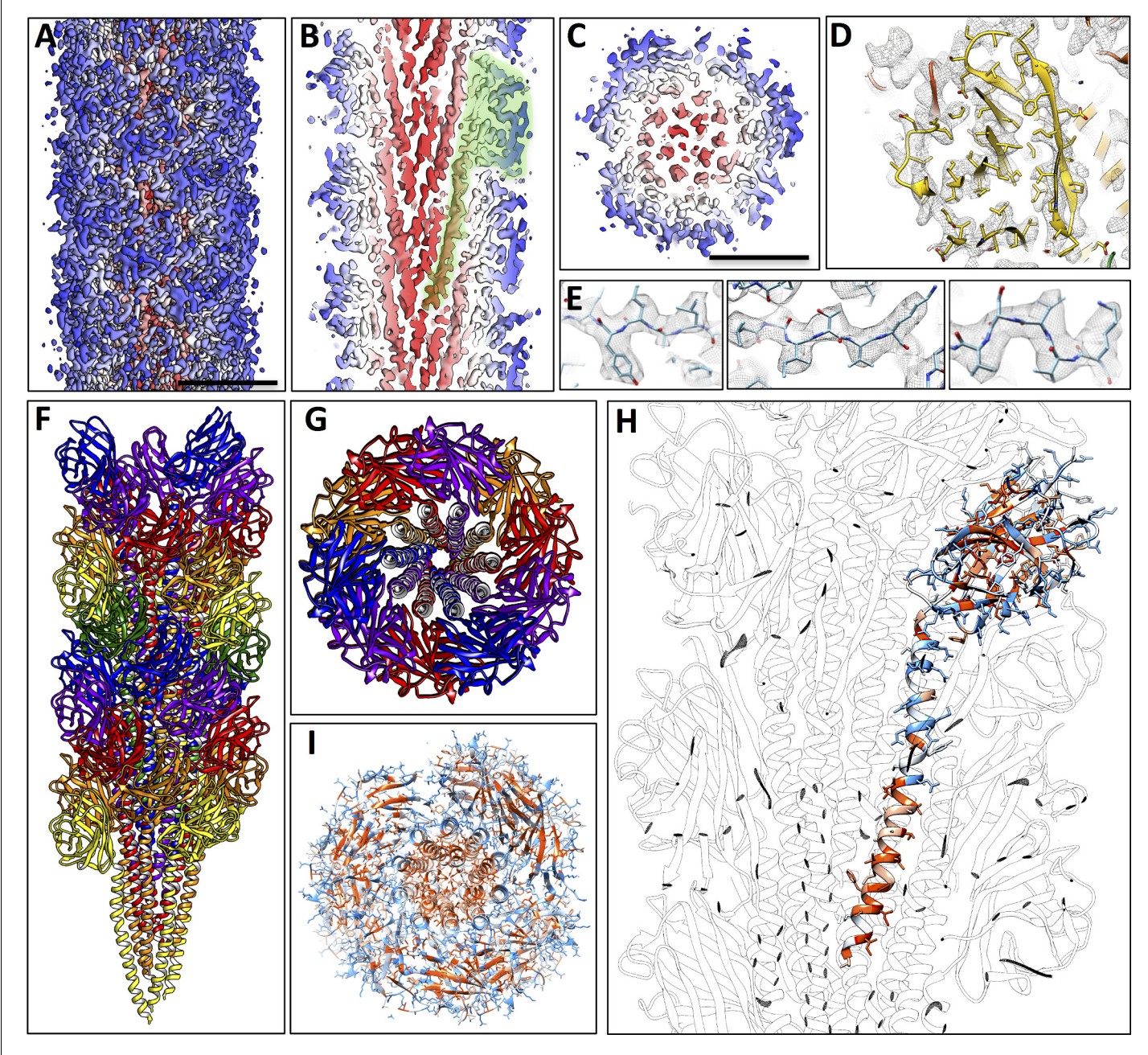

**Figure 5.** Structure of the *P. furiosus* archaellum. (A–C) 3D representation of the 4.2 Å map of the *P. furiosus* archaellum as seen from the surface (A), and cross-sections parallel (B) and perpendicular (C) to the long axis of the filament. Different colours represent different regions of the archaellum; red, inner helix bundle; white – blue, outer beta-strand-rich sheath; transparent green, outline of one archaellin monomer. Scale bars A, C, 50 Å. (D) Slice through the outer sheath of the filament showing the $\beta$-strand rich region of the $FlaB_0$ monomer (yellow) fitted into the map density (transparent grey). Note hydrophobic amino acid side chains pack in the interior of the $\beta$-barrel. (E) Close-ups of beta strands of $FlaB_0$ (backbone in blue) fitted into the map density (transparent grey). (F, G) Side view (F) and cross-section (G) of the atomic model of the *P. furiosus* archaellum with individual $FlaB_0$ subunits in different colours. (H) structure of the $FlaB_0$ monomer coloured by hydrophobicity (red, hydrophobic; blue, hydrophilic). Neighbouring subunits within the filament are shown in transparent grey. (I) structure of the *P. furiosus* archaellum coloured by hydrophobicity in top view.

The following figure supplements are available for figure 5:

**Figure supplement 1.** Resolution estimation and model validation of the *P. furiosus* archaellum.

*Figure 5 continued on next page*

*Figure 5 continued*

**Figure supplement 2.** Multiple sequence alignment between *P. furiosus* FlaB$_0$, FlaB$_1$ and FlaB$_2$ using the Praline server (http://www.ibi.vu.nl/programs/pralinewww/), showing sequence conservation.

**Figure supplement 3.** Multiple sequence alignment of *P. furiosus* FlaB$_0$, FlaB$_1$ and FlaB$_2$ using the Praline server (http://www.ibi.vu.nl/programs/pralinewww/), showing secondary structure prediction (helices, red; beta strands, blue).

**Figure supplement 4.** Transmembrane helix prediction of *P. furiosus* FlaB$_0$ using the TMHMM server (http://www.cbs.dtu.dk/services/TMHMM/) predicting residues 1–6 inside, 7–29 as transmembrane helix and 30–212 outside (periplasm).

**Figure supplement 5.** Comparison between three archaeal filaments.

**Figure supplement 6.** Sequence alignment of *P. furiosus* FlaB$_0$, *M. hungatei* FlaB$_3$ and the *I. hospitalis* 670 polypeptides.

(*Figure 5H–I*). Similar to bacterial T4P-like filaments, (*Giltner et al., 2012*; *Pelicic, 2008*), the amphipathic character of FlaB$_0$ would provide the driving force for filament assembly, in which hydrophobic domains are buried in the centre and hydrophilic surfaces exposed to the surrounding medium. In line with this, it has been shown that archaellin monomers from denatured archaella can spontaneously reassemble into filamentous structures (*Näther-Schindler et al., 2014*).

Comparing the structure of the *P. furiosus* archaellum with that of the *Methanospirillum hungatei* archaellum (*Poweleit et al., 2016*) as well as the adhesion fibre from the immotile *Ignicoccus hospitalis* (*Braun et al., 2016*; *Müller et al., 2009*) (*Figure 5—figure supplement 5A,B*), highlights different degrees of domain-specific conservation. Both archaellins (*P. furiosus* PfuFlaB$_0$ and *M. hungatei* MhuFlaB$_3$), as well as the *I. hospitalis* (Iho670) fibre subunit are structurally conserved in the N-terminal helix (*Figure 5—figure supplements 6*), which in assembly forms the backbone of each filament (*Figure 5—figure supplement 5A,B*). In addition, the three proteins show the same amphipathic

**Table 1.** Statistics of 3D reconstruction and model refinement.

| Data collection | |
|---|---|
| Electron microscope | JEOL JEM 3200 FSC |
| Electron detector | K2 in counting mode |
| Voltage | 300 kV |
| Defocus range | 1–3 μm |
| Pixel size | 1.12 Å |
| Electron dose | 60 e$^-$/Å$^2$ |
| Images | 297 |
| **3D reconstruction** | |
| Final particles | 13,965 helical segments |
| Resolution | 4.2 Å |
| *B* factor | −200 Å$^2$ |
| **Ramachandran plot** | |
| Favored | 82.50% |
| Outliers | 0% |
| **Validation** | |
| EMringer score | 2.06 (*Barad et al., 2015*) |
| MolProbity score | 2.54 |
| Rotamer favored | 92.26% |
| Rotamer outliers | 1.64% |

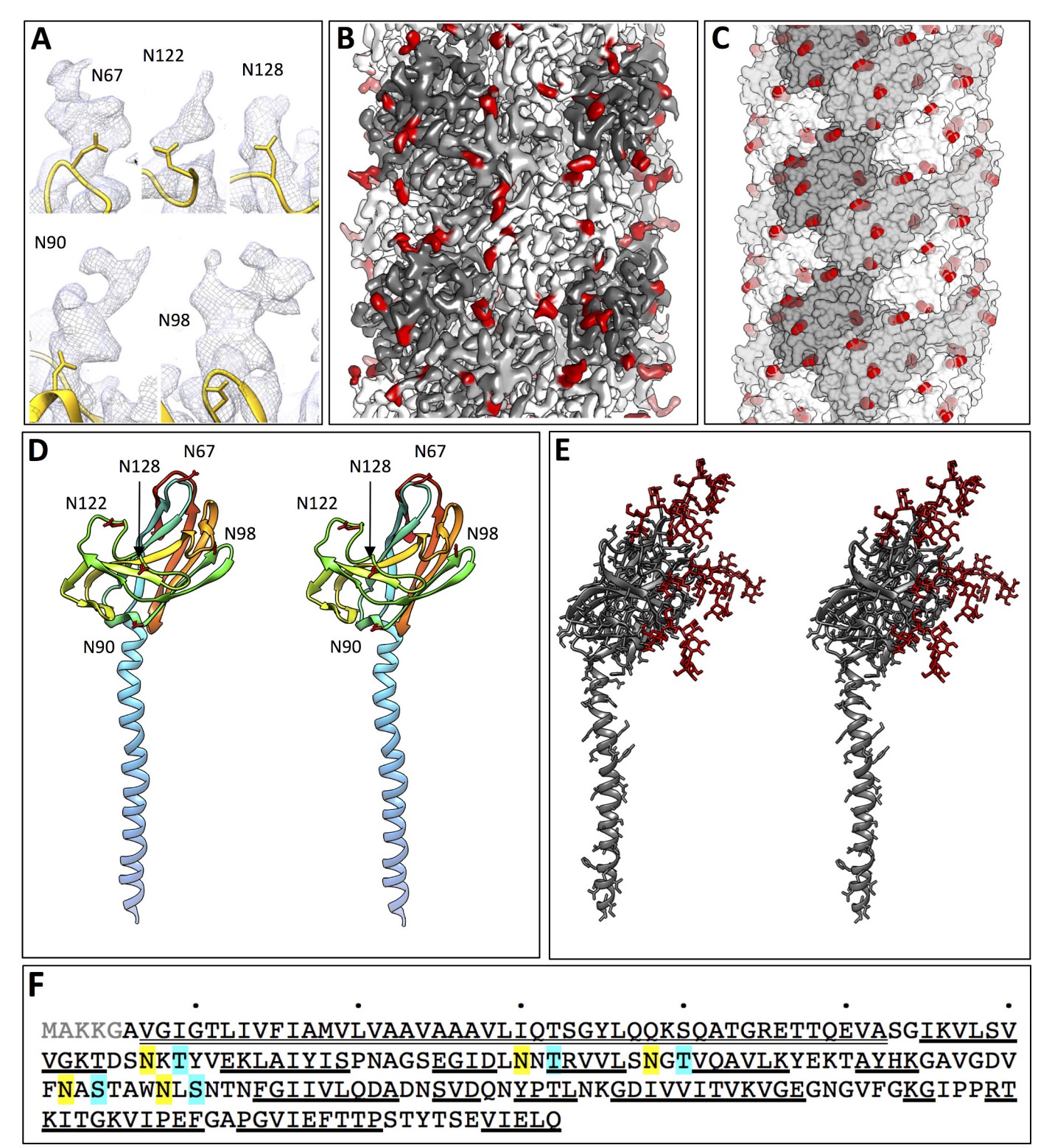

**Figure 6.** Glycosylation of the *P. furiosus* archaellum. (A) Close-ups of glycan densities near Asn residues. (B) surface representation of EM map showing glycan densities (red) protruding from the filament (shades of grey). (C) Surface representation of the atomic model of the archaellum (shades of grey, individual FlaB$_0$ subunits; red, asparagine residues within glycosylation sequon). (D) stereo view of the *P. furiosus* FlaB$_0$ monomer in rainbow representation (blue, N-terminus; red, C-terminus) with glycosylated asparagines labelled in red. (E) stereo view of the *P. furiosus* FlaB$_0$ monomer (dark

*Figure 6 continued on next page*

*Figure 6 continued*

grey) and glycan structures (red) modelled near glycosylated Asn residues. (F) sequence of *P. furiosus* FlaB$_0$. Grey, clipped signal peptide; double line, $\alpha$-helix; single line, $\beta$ strand; yellow, glycosylated asparagine; blue, T/S residue in conserved glycosylation sequon; every 10$^{th}$ residue of the *P. furiosus* sequence labelled by a dot.

character (*Figure 5—figure supplement 5E*) and assemble into filaments that have similar helical parameters but distinct diameters (100 Å for *M. hungatei,* 110 Å for *P. furiosus* and 158 Å for *I hospitalis; Figure 5—figure supplement 5 and A,B*). These differences in filament thickness are due to insertions and deletions in the C-terminal beta strand-rich domains of *Pfu*FlaB$_0$, *Mhu*FlaB$_3$ and *Iho*670 (*Figure 5—figure supplements 6*), which most likely reflect functional and environmental adaptations in the three species.

## N-glycosylation of the *P. furiosus* archaellum

While building the atomic FlaB$_0$ model, we found large, irregular, outward-facing densities adjacent to five out of 13 asparagine residues (N67, N90, N98, N122 and N128) (*Figure 6A*), which were not attributable to the polypeptide backbone or nearby sidechains. Analysis of the FlaB$_0$ sequence revealed that these five amino acids were all within a N-X-S/T sequon (*Figure 6F*), a highly conserved consensus sequence for N-linked glycosylation (*Jarrell et al., 2014*). Consistent with this finding, the remaining asparagine residues were not part of such a sequon and did not show a similar density nearby. Furthermore, no such densities were observed adjacent to serine, threonine, proline or lysine sidechains, suggesting that O-linked glycosylation is not present in the *P. furiosus* archaellum. Due to the flexibility of glycan chains, only the densities of the first one or two sugar subunits were present in our map. The *N*-glycosylation sites are evenly distributed over the exposed part of the iFlaB$_0$ peptide (*Figure 6D*), thus creating a homogeneous, highly glycosylated surface that covers the entire filament (*Figure 6B,C*).

The universal sequence of the *P. furiosus* N-glycan has been determined previously by mass spectrometry (*Fujinami et al., 2014*). To analyse the additional molecular density corresponding to FlaB$_0$ glycosylation, we generated an atomic model of the *P. furiosus* glycan and placed it adjacent to each of the glycosylated asparagine residues (*Figure 6E*). The resulting model shows that glycosylation adds a substantial mass to each monomeric unit and thus covers most of the filament surface. This glycosylation pattern differs considerably from the one shown for the archaellum of *M. hungatei* (*Poweleit et al., 2016*), which harbours six O-glycosylation sites and two N-glycosylated residues. Although glycosylation has not yet been shown for the *I. hospitalis* filament, one sequon for N-glycosylation can be found within the polypeptide, suggesting a singular putative 'sweet spot' (*Figure 5—figure supplement 6*).

## Discussion

### Assembly of the archaellum machinery

Together with a wealth of previous biochemical data, our new structures of the *Pyrococcus furiosus* archaellar motor complex, the polar cap, the S-layer, and the archaellum itself enable us to build a first model describing the architecture of the archaellum machinery (*Figure 7*).

Filament assembly and rotation is powered by the archaellar motor, which is composed of the fully membrane-embedded FlaJ, a bell-shaped cytosolic complex of FlaI and FlaH and a surrounding cytosolic ring, most likely consisting of FlaC and D/E. This ring was not observed in averages of the archaellar motor complex of *Thermococcus kodakarensis* (*Briegel et al., 2017*), which may either be due to species-dependent structural differences of motor components or the smaller number of particles used for the *T. kodakarensis* average. In the periplasm, the filament is thought to be coordinated by FlaF (*Banerjee et al., 2015*). As no densities spanning the entire periplasm are seen in our sub-tomogram average of the motor complex (*Figure 2A*), we suggest that FlaF does not form a continuous conduit that connects the membrane and the outer canopy of the S-layer. Instead, this protein may coordinate the archaellum near the membrane, while the main periplasmic part of the filament is flexibly integrated into the S-layer.

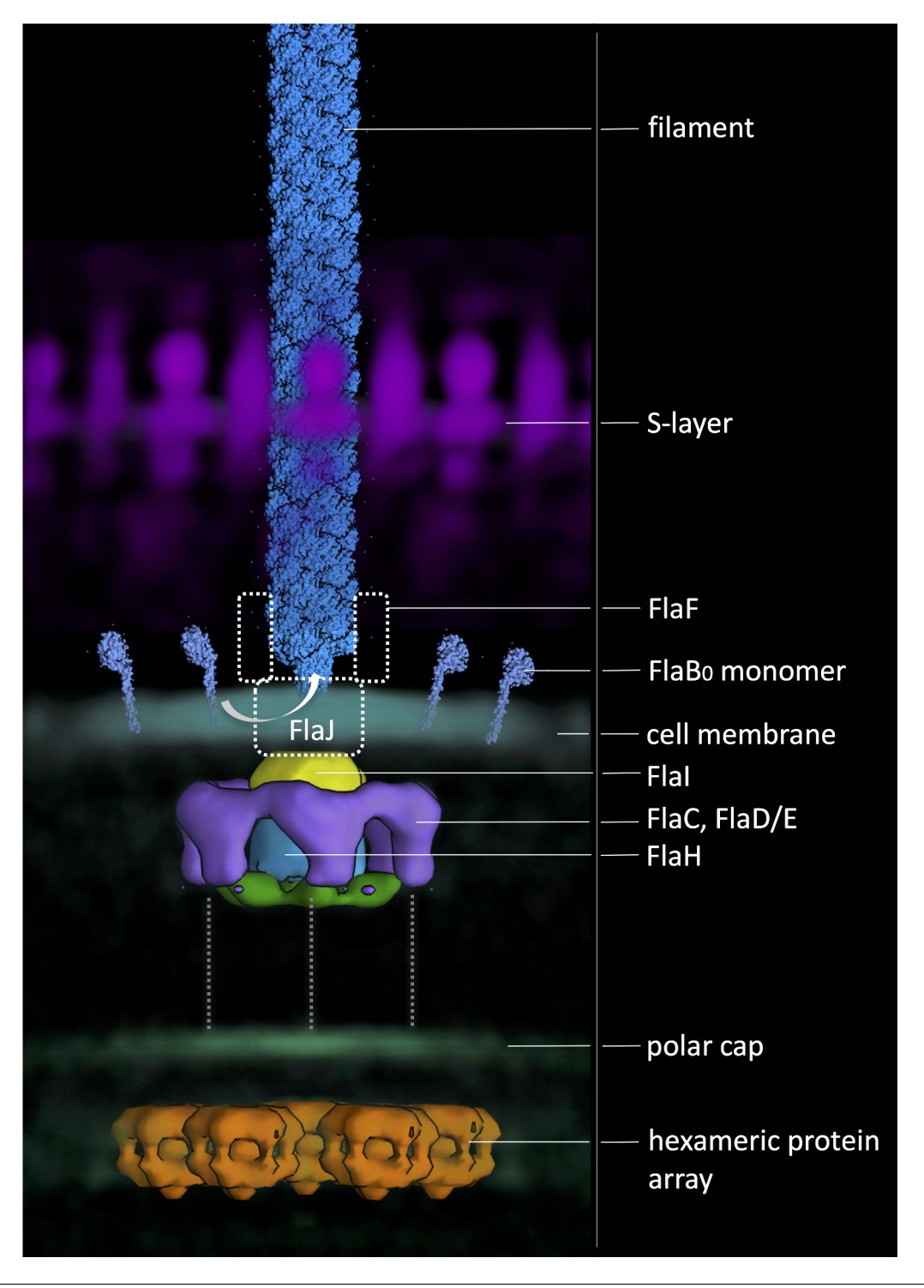

**Figure 7.** Composite model of the archaellum machinery of *P. furiosus*. Light blue, FlaB$_0$ monomers and filament (from helical reconstruction); hazy magenta, S-layer; solid yellow, blue, green and purple, motor complex; hazy blue, cell membrane; hazy green, polar cap; solid orange, hexagonal protein array (from different sub-tomogram averages). Putative positions of protein subunits are indicated. Dashed grey lines, putative interaction with polar cap.

The archaellar motors are juxtaposed to the polar cap, which in *P. furiosus* is only present in combination with an archaellar bundle. This confirms earlier findings in *H. salinarum* (*Kupper et al., 1994*) and *T. kodakarensis* (*Briegel et al., 2017*), which also showed co-localisation of archaella and polar cap. This striking co-localisation indicates that archaella and polar cap are co-regulated and suggests a strong functional connection. It is conceivable that the polar cap functions in concentrating archaella at one cell pole and acts as an anchor to fix the motor complexes in the bilayer to prevent futile rotation. In addition, the polar cap may also prevent the mechanical disruption of the cytoplasmic membrane by the rapid rotation of multiple archaella.

The straight sides and tightly bent edges of the polar cap are reminiscent of virus-associated pyramids, the exotic egress structures produced by *Sulfolobus*-specific viruses upon infection of their hosts (*Daum et al., 2014*). These observations strongly suggest that the polar cap is composed of protein, rather than lipid. In addition, the polar cap is associated with protein complexes that form hexameric protein arrays. While the identity of these complexes is unknown, their localisation next to the archaellar motors suggests that they may be mechanistically linked to motor function. Many bacteria employ chemoreceptors to sense metabolic gradients and control the direction of swimming motion. These chemoreceptors are typically arranged in hexagonal arrays (*Briegel et al., 2009*; *Liu et al., 2012*; *Qin et al., 2016*) and closely associated with flagellar motors (*Liu et al., 2012*; *Qin et al., 2016*; *Xu et al., 2011*). In Euryarchaeota, genes homologous to bacterial chemoreceptors have been annotated (*Schlesner et al., 2009*) and chemosensory arrays that are structurally similar to their bacterial counterparts have been described (*Briegel et al., 2015*). It remains to be investigated if the hexagonal protein arrays found in this study represent an unknown *Pyrococcus*-specific type of chemoreceptor or some other protein linked to motility. Our observations support the hypothesis that the polar cap does not only function as an anchor for the archaellum, but also as a platform that recruits protein complexes that control swimming motion (*Briegel et al., 2017*).

The *P. furiosus* archaellum lacks a central channel, consistent with the archaellum of *M. hungatei* (*Poweleit et al., 2016*) and the *I. hospitalis* Iho670 filament (*Braun et al., 2016*) *Figure 5—figure supplement 5A*. This supports the hypothesis that archaella and other archaeal filaments assemble from their base, in marked contrast to the assembly of bacterial flagella (*Evans et al., 2014*).

The first steps of archaellum assembly comprise N- or O-glycosylation and removal of the positively charged N-terminal MAKKG signal peptide from membrane-bound $FlaB_0$ monomers. Loss of these positive charges primes the individual archaellins for transfer from the lipid bilayer into the growing filament, aided by the membrane protein FlaJ. This process is driven by the amphipathic surface of the individual archaellins and catalysed by ATP hydrolysis through FlaI (*Streif et al., 2008*; *Reindl et al., 2013*). In contrast to the *M. hungatei* filament (*Poweleit et al., 2016*), no O-linked glycosylation was found in *P. furiosus*, indicating that this glycosylation pathway is not present in this organism.

The extensive glycan cover of the archaellum filaments likely increases their stability. All but one of the glycosylation sites are situated near the binding interface of adjacent $FlaB_0$ subunits (*Figure 6C,D*), suggesting that N-linked glycans may enhance filament integrity by additional hydrogen bonds (*Lis and Sharon, 1993*). Indeed, deletion mutants of different archaeal species lacking important proteins involved in glycosylation pathways are defective in archaellum formation and motility (*Jarrell et al., 2014*). A high degree of surface protein N-glycosylation has been suggested to be a protection mechanism against heat in the case of (hyper-) thermophiles, or against high pH and salt concentrations in other archaeal extremophiles (*Jarrell et al., 2014*). In addition, glycosylation of proteins may convey adhesive properties (*Lis and Sharon, 1993*), resulting in the surface-binding capability of archaella previously observed for adherent *P. furiosus* cells (*Näther et al., 2006*). Finally, surface glycans also provide strain-specific recognition signatures for cell-cell interactions, in which archaella seem to play a major role (*Näther et al., 2006*).

In its substance, our model of the archaellum and its motor complex will be universal to all motile Archaea, as the core of the archaellum machinery (FlaA/B, FlaF, FlaG, FlaH, FlaI and FlaJ) is conserved throughout crenarchaeal, euryarchaeal and thaumarchaeal lineages (*Albers and Jarrell, 2015*; *Chaudhury et al., 2016*). In contrast, it can be expected that the cytosolic ring will be different in Crenarchaea, which lack genes encoding for FlaC D/E but possess the ring-forming protein FlaX instead (*Banerjee et al., 2012*). Moreover, together with previous findings (*Briegel et al., 2017*; *Kupper et al., 1994*) our data suggest that the polar cap is a unique hallmark of Euryarchaea.

## Evolutionary aspects

Archaeal filament proteins such as the archaellins $Pfu$FlaB$_0$ and $Mhu$FlaB$_3$, as well as the adhesive protein $Iho$670 share many characteristics with T4 pilins. These characteristics include their overall lollipop-like shape, their amphipathic surfaces and the way they are processed and assembled into a filament. On the sequence level, archaellins and T4 pilins are highly conserved in their N-terminal helix and unconserved in their globular, $\beta$-strand-rich domain (*Poweleit et al., 2016*; *Giltner et al., 2012*; *Pelicic, 2008*). It is likely that the N-terminal helix domain remained largely unaltered throughout evolution, as it is the essential backbone-forming domain in T4P and archaella. In contrast, the surface-exposed globular domain is less crucial for filament integrity and could therefore diversify into a versatile and strain-specific interaction interface that adapted to various functions and environments. An extreme example of this diversification are the class III signal peptide-containing sugar binding proteins of the bindosome of *Sulfolobus solfataricus* (*Zolghadr et al., 2007*).

The ubiquity and conservation of the archaellin/T4 pilin (A/T) blueprint highlights its evolutionary success as an exquisite building block for stable, yet flexible and versatile filaments. The presence of this blueprint in Archaea and Bacteria also raises the intriguing hypothesis that a common A/T-like progenitor protein and filament existed in the last universal common ancestor (LUCA), before these two domains of life diverged more than 3 billion years ago (*Makarova et al., 2016*; *Weiss et al., 2016*). Although it is unknown if these primordial filaments were used for rotary propulsion, it has been proposed that pseudopili of T2SS, an evolutionary relative of the archaellar motor, assemble in a rotary 'spool-like' fashion (*Nivaskumar et al., 2014*). This suggests that the ability to gyrate is a general trait of T2SS as well as A/T-like assembly systems. Thus, Archaea appear to have developed the A/T progenitor filament and machinery further into the archaellum, a structurally simple, yet powerful propulsion device. Bacteria on the other hand, have developed this ancestral fibre into T4P, molecular devices used for processes such as twitching motility as well as DNA uptake (*Gold et al., 2015*). Notably, the architecture of the bacterial T4P assembly machinery differs greatly from the archaellar motor. Whereas the archaellar motor apparently lacks a conduit through the periplasm, the bacterial T4P machinery harbours the PilQ complex, a large, gated molecular machinery that guides the pilus through the peptidoglycan layer and the outer membrane (*Gold et al., 2015*; *Chang et al., 2017*, *2016*; *Kolappan et al., 2016*).

For swimming motion, Bacteria have developed the flagellar machinery, a massive double-membrane spanning macromolecular device that is thought to share a common ancestor with the bacterial type-3 secretion system (*Chen et al., 2011*; *Erhardt et al., 2010*). The evolutionary reason for the higher complexity of both, bacterial T4P assembly machines as well as flagellar motors is speculative. However, it may be hypothesised that Bacteria needed to come up with additional protein components for their T4P assembly machinery as well as an entirely different propulsion device as they developed a double membrane, which is, with very few exceptions, not found in Archaea.

# Materials and methods

## Cultivation of cells

*Pyrococcus furiosus* type strain DSM3638 was obtained from the in-house culture collection at the University of Regensburg. Cells were grown in serum bottles containing 20 ml minimal SME pyruvate medium (= modified SME medium supplemented with 0.025% yeast extract and 40 mM pyruvate), pressurized with 100 kPa N$_2$ (*Reichelt et al., 2016*). Incubation took place overnight at 95°C.

## Electron cryo-tomography

Cells were decanted from serum bottles into 15 ml tubes and pelleted at 2500 g. The cell pellet was resuspended in 2 pellet volumes of buffer and mixed 1:1 with 10 nm colloidal protein-A gold (Aurion, Wageningen, The Netherlands). 3 μl of this suspension were applied to glow-discharged 300 mesh copper Quantifoil grids (R2/2, Quantifoil, Jena, Germany), blotted for 3–5 s and rapidly injected into liquid ethane using a homemade plunge-freezer.

Tomograms were recorded using a Polara G2 Tecnai TEM (FEI, Hillsboro, USA) operating at 300 kV. The Polara was equipped with a Gatan Tridiem energy filter (Gatan Inc., Pleasanton, USA) and a $4 \times 4$ k K2 Summit direct electron detector (Gatan Inc., Pleasanton, USA) running in counting mode. Tilt series were collected in zero-loss mode using Digital Micrograph (Gatan, Pleasonton, USA) from

max. −68° to +68° and in steps of 2°. The magnification was set to 41,000 x resulting in a final pixel size of 5.4 Å, and a defocus of 6–8 μm was applied. Tomograms were recorded in dose-fractionation mode at a dose rate of 8–10 e$^-$ px$^{-1}$ s$^{-1}$ and a maximum total dose of 100 e$^-$ Å$^{-2}$. Dose-fractionated tilt images were aligned using an in-house script based on IMOD (*Kremer et al., 1996*) programmes and reconstructed into tomograms using the IMOD software package (*Kremer et al., 1996*). Final tomograms were binned 2-fold and contrast-enhanced using non-linear anisotropic diffusion (*Frangakis and Hegerl, 2001*). 3D surface annotation (segmentation) was performed using AMIRA (FEI, Hillsboro, USA).

## Sub-tomogram averaging of the motor complex

To obtain averages of motor complexes, 379 individual sub-volumes were picked from 50 tomograms binned twofold to a pixel size of 10.8 Å and filtered by NAD (*Frangakis and Hegerl, 2001*) as implemented in IMOD (*Kremer et al., 1996*). For each particle, two coordinates were selected, of which the first marked the centre of the periplasm and the second marked the central part of the motor density. Two of three Euler angles (Psi and Theta) were roughly determined by the two contours. Particles were then pre-aligned and averaged using IMOD (*Kremer et al., 1996*) and Spider (*Shaikh et al., 2008*) as previously described (*Davies et al., 2011*). This rough average was used as initial reference for PEET (*Nicastro et al., 2006*), in which particle alignment was refined. 6-fold symmetry was applied by duplicating and rotating particles by 0, 60, 120, 180, 240 and 300° around the symmetry axis. The membrane and surrounding noise were excluded by a cylindrical mask around the motor complex. To increase map resolution, averages were recreated without additional alignment search, using unfiltered tomograms binned twofold to a pixel size of 10.8 Å. Fourier shell correlation (FSC) of two particle half-sets as implemented in IMOD (*Kremer et al., 1996*) was used to estimate the resolution of the resulting map at the 0.5 criterion. For this, the central particle was masked using a box of x,y,z = 30,30,30 nm. This analysis revealed an estimated resolution of 68 Å (*Figure 2—figure supplement 2*). To analyse the 3D organisation of motor complexes *in situ*, averaged particles were placed back into their relative positions and orientations using the EM-Package in Amira (FEI, Hillsboro, USA).

## Sub-tomogram averaging of the S-layer and the hexagonal protein array

To average the S-layer, a random grid of 1951 points was applied over the S-layer within a tomogram of a *P. furiosus* cell that was binned twofold to a pixel size of 10.8 Å and filtered by NAD (*Frangakis and Hegerl, 2001*). Using PEET (*Nicastro et al., 2006*), subvolumes of x,y,z = 60,60,40 pixels were extracted, aligned and averaged. To average the accessory protein array, 57 subvolumes were selected manually using a tomogram of a *P. furiosus* cell that was binned 2-fold to a pixel size of 10.8 Å and filtered by NAD (*Frangakis and Hegerl, 2001*). Using PEET (*Nicastro et al., 2006*), subvolumes of 80 × 80 × 80 pixels were extracted, aligned and averaged applying 6-fold symmetry. S-layer and protein array were visualised and segmented using UCSF Chimera (*Pettersen et al., 2004*). The resolution of both averages was estimated based on reflections in their respective power spectra calculated by IMOD (*Kremer et al., 1996*). This suggested a resolution of ~52 Å for the average of the S-layer (*Figure 3—figure supplement 1*) and ~49 Å for the hexagonal protein array (*Figure 4—figure supplement 1*).

## Preparation of archaella

Preparation of archaella was adapted from *Näther et al. (2006)*. *P. furiosus* was grown in modified SME medium supplemented with 0.1% yeast extract, 0.1% starch, and 0.1% peptone at 95°C in a 50 l fermenter (Bioengineering, Wald, Switzerland) pressurized with 100 kPa N$_2$/CO$_2$ (80:20). After harvesting (7,000–8,000 × g), cells were concentrated (3,500 × g, 30 min, 4°C; Sorvall RC 5C plus, rotor GS3), and archaella were sheared (Ultraturrax T25, IKA-Werke, Staufen, Germany; 1 min at 13,000 rpm and 10 s at 22,000 rpm). Cell debris was removed by centrifugation (34,500 × g, 4°C; Sorvall RC 5C plus, rotor SS34) for 20 min. Archaella were pelleted from the supernatant by ultracentrifugation (60,000 × g, 90 min, 4°C; Beckman Optima LE-80K, rotor 70Ti), resuspended in 150 μl 0.1 M HEPES pH 7.0 and purified for 48 hr (250,000 × g, 4°C; Beckman Optima LE-80K, rotor SW60-Ti) using a CsCl gradient (0.45 g/ml). Fractions were taken by puncturing the ultracentrifuge tubes with sterile

syringes and dialysed against aerobic ½ SME/5 mM HEPES pH 7.0. After identification of the arch-aella-containing band by SDS-PAGE and TEM, the respective sample was stored at 4°C for further analysis.

### Expression and purification of *P. furiosus* FlaI (*Pf* FlaI) from *E. coli*

Overexpression constructs of StrepII-tagged *Pf* FlaI (*Table 2*) were transformed into *E. coli* codon plus (Rosetta) cells and grown as preculture overnight at 37°C in LB medium containing ampicillin (50 µg/ml) and chloramphenicol (34 µg/ml). 2 l of fresh medium containing antibiotics were inoculated with 10 ml preculture and grown at 37°C to an $OD_{600}$ of 0.6–0.7. Subsequently, the cultures were cooled down on ice for 30 min and induced with 0.3 mM of isopropyl β-D-thiogalactopyranoside. Growth was continued for 16 hr at 18°C. Cells were collected by centrifugation, frozen in liquid nitrogen and stored at −80°C.

Cells expressing *Pf* FlaI were thawed on ice and resuspended in lysis buffer (20 mM Tris-HCl pH 8, 150 mM NaCl, 0.5% TritonX100) containing complete EDTA-free protease inhibitor cocktail (Roche) (5 ml/g of pellet). DNase I was added (a pinch) and kept on ice for 30 min. The cells were lysed using a Microfluidizer with 1000 psi three times and centrifuged at 4,600 g for 20 min to remove cell debris and then centrifuged at 20,000 g for 20 min at 4°C. For affinity chromatography, Streptactin column material (IBA GmbH, Göttingen, Germany) was prepared in columns equilibrated with purification buffer, 20 mM Tris/HCl pH 8, 150 mM NaCl (buffer A) and proteins were eluted in buffer A containing 2.5 mM desthiobiotin.

The protein was purified further by size exclusion chromatography, where *Pf* FlaI was concentrated to 1 mg/ml in buffer A using 10 kDa cutoff Amicon concentrators (Millipore). 500 µl of the concentrated samples was then applied to a Superdex 200 10/300 GL size exclusion column equilibrated with buffer A. Fractions were analysed on SDS-PAGE. Thyroglobulin (669 kDa), γ globulin (158 kDa), ovalbumin (44 kDa), myoglobin (17 kDa) and vitamin B12 (1.35 kDa) were used as size standards.

### Negative-stain EM and single-particle analysis of FlaI

Purified FlaI was diluted 1:5 in buffer A and 3 µl of this suspension were added to 300 or 400 mesh continuous-carbon copper grids. After 1 min incubation, the suspension was washed 1 x with 3 µl buffer and subsequently 3 µl uranyl acetate stain were added. After incubation for another minute, excess uranyl acetate was removed and grids were transferred into a FEI Tecnai Spirit TEM (Eindhoven, The Netherlands) operating at 120 kV. Images were recorded on a Gatan US 4000 camera (Gatan Inc., Pleasanton, USA) at a pixel size of 1.4 Å and a defocus range of 1–3 µm using the LEGINON software (*Suloway et al., 2005*). 2D classification of 130,000 auto-picked particles was performed using the software Relion-2.0 (*Kimanius et al., 2016*).

### CryoEM of archaella and helical processing

Isolated *P. furiosus* archaella were diluted 1:5 in Millipore water and 3 µl of the suspension was applied to glow-discharged 300 mesh copper Quantifoil grids (R2/2, Quantifoil, Jena, Germany). The grids were plunge-frozen in liquid ethane with a Vitrobot III (FEI, Eindhoven, The Netherlands) using a blotting time of 7–10 s and transferred at liquid nitrogen temperature into a JEOL JEM 3200 FSC operating at 300 kV. Dose-fractionated movies were collected using a K2-Summit detector running in counting mode. The slit width of the in-column energy filter was set to 20 eV. 40-frame movies were recorded at a defocus of 1–3 µm with an exposure time of 8 s and a total dose of 60 e⁻/Å². 297 movies were drift-corrected and dose-filtered using Unblur (*Grant and Grigorieff, 2015*) and

**Table 2.** Plasmid used in this study.

| Plasmid | Relevant characteristics | Source |
|---|---|---|
| pSVA 3116 | pETDuet-1 carrying N-terminal His6 tagged *Pf*FlaI, Amp[R]. | (*Chaudhury et al., 2016*) |
| pSVA 3140 | pETDuet-1 containing N-terminal StrepII-tagged *Pf*FlaI. The PCR product obtained using the 5118 and 5110 primers (see *Table 3*) on pSVA3116 was cloned using the NcoI and PstI sites. Amp[R] | This study |

**Table 3.** Primers used in this study. Relevant restriction sites are underlined.

| Primers | Sequence and characteristics | Source |
|---------|------------------------------|--------|
| 5110 | 5'-GGG<u>CTGCAG</u>TCAGATTCTGAAGCTTAGTC-3' | (*Chaudhury et al., 2016*) |
| 5118 | 5'GGG<u>CCATGG</u>GCTGGAGTCATCCACAATTTGAGAAGATGGCGGAAGTTATGTCAC-3' | This study |

4712 helices were manually picked using the programme Helixboxer of the EMAN-2 package (*Ludtke et al., 1999*). Subsequent image processing included CTF-correction, particle extraction, 3D classification, 3D refinement and B-factor sharpening and was performed in Relion-2.0 (*He and Scheres, 2017*). In brief, each filament was subdivided into boxes of 200 × 200 pixels in size at an offset of 10 Å, resulting in 74,823 overlapping segments. Next, a 3D consensus map of all extracted particles was calculated, by performing a narrow-range angular and translational search around the helical parameters published for the *M. hungatei* archaellum (*Poweleit et al., 2016*). The resulting consensus map was used as reference in a 3D classification step to sort out bad, bent or misaligned filaments. The best class including 13,965 helical segments was subjected to another round of 3D refinement in which helical parameters were further improved. This resulted in a map with a resolution of 4.2 Å by gold-standard FSC (*Figure 5—figure supplement 1*). In a final post-processing step, the map was sharpened using a B-factor of $-200$ Å$^2$. Statistics of 3D reconstruction and model refinement can be found in *Table 1*.

### Structure analysis, model building and sequence alignment

Sub-tomogram averages as well as the archaellum structure were displayed and analysed in UCSF Chimera (*Pettersen et al., 2004*). The same programme was used to perform rigid body fitting of X-ray structures of *S. acidocaldarius* FlaI (PDB-4IHQ) and FlaH (PDB-4YDS) into the sub-tomogram average. SymmDock (*Schneidman-Duhovny et al., 2005*) was employed to generate multimeric rings from the *S. acidocaldarius* FlaH monomer.

The atomic model of the archaellum was built in Coot (*Emsley et al., 2010*) aided by homologous *M. hungatei* FlaB$_3$ (PDB 5tfy). Non-homologous features were built manually. The structure was refined by Phenix (*Afonine et al., 2012*) real space refinement followed by manual rebuilding in Coot. The final model contains all 207 amino acid residues of the mature protein (*Pettersen et al., 2004*). Multiple sequence alignments were calculated using the Praline server (http://www.ibi.vu.nl/programs/pralinewww/). The sequence for the *P. furiosus* glycan was converted into a molecular model using the SWEET2-server http://www.glycosciences.de/modeling/sweet2/doc/index.php.

### Data deposition

The cryo-EM maps were deposited in the Electron Microscopy Data Bank with accession codes EMD-3759 (motor complex), EMD-3760 (hexagonal protein array) and EMD-3746 (archaellum). The structure coordinates of the atomic model of the archaellum were deposited in the Protein Data Bank with accession number 5O4U.

## Acknowledgements

We thank Deryck Mills for keeping our electron microscopes in perfect working order, Edoardo D'Imprima for advice on single-particle EM, Carsten Sachse for guidance on helical reconstruction and Vicki Gold and Harald Huber for many inspiring discussions. We also acknowledge Thomas Hader, Konrad Eichinger and Dina Grohmann for supporting the mass cultivation of *P. furiosus* cells. This project was funded by the Max Planck Society (BD, JV, WK), the University of Exeter Research Fellow's Startup grant (BD), the ERC starting grant 'ARCHAELLUM' (511323; SVA) and the University of Regensburg (ReR, RaR, AB).

## Additional information

### Competing interests

WK: Reviewing editor, *eLife*. The other authors declare that no competing interests exist.

### Funding

| Funder | Grant reference number | Author |
| --- | --- | --- |
| Max-Planck-Gesellschaft | research funding | Bertram Daum<br>Janet Vonck<br>Werner Kühlbrandt |
| University of Exeter | Research Fellow Starting Grant and open access funding | Bertram Daum |
| Universität Regensburg | research funding | Annett Bellack<br>Robert Reichelt<br>Reinhard Rachel |
| European Commission | Archaellum Project ID: 311523 | Paushali Chaudhury<br>Sonja-Verena Albers |

The funders had no role in study design, data collection and interpretation, or the decision to submit the work for publication.

### Author contributions

BD, Conceptualization, Investigation, Writing—original draft, Writing—review and editing; JV, Data curation, Formal analysis, Validation, Investigation, Visualization, Writing—original draft; AB, Resources, Methodology, Writing—original draft; PC, RRe, Resources, Methodology; S-VA, Resources, Supervision, Funding acquisition, Methodology, Writing—original draft; RRa, Conceptualization, Methodology, Writing—original draft; WK, Supervision, Funding acquisition, Writing—original draft, Writing—review and editing

### Author ORCIDs

Bertram Daum, http://orcid.org/0000-0002-3767-264X

Janet Vonck, http://orcid.org/0000-0001-5659-8863

Annett Bellack, http://orcid.org/0000-0001-9771-0348

Sonja-Verena Albers, http://orcid.org/0000-0003-2459-2226

Reinhard Rachel, http://orcid.org/0000-0001-6367-1221

Werner Kühlbrandt, http://orcid.org/0000-0002-2013-4810

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
