## [Decision Letter]

Thank you for submitting your article "Structure and *in situ* organisation of the *Pyrococcus furiosus* archaellum machinery" for consideration by *eLife*. Your article has been favorably evaluated by John Kuriyan (Senior Editor) and three reviewers, one of whom is a member of our Board of Reviewing Editors. The reviewers have opted to remain anonymous.

The reviewers have discussed the reviews with one another and the Reviewing Editor has drafted this decision to help you prepare a revised submission.

Summary:

This manuscript describes structural studies of the *Pyrococcus furiosus* flagellar machinery. Using a combination of electron cryo-tomography, sub-tomogram averaging, single-particle cryoEM and helical reconstruction, the authors generated a partial model of the machinery. The main concerns raised were that portions of the model are quite speculative given the very low resolution and incompleteness, while the higher resolution component (the flagellar filament) lacks novelty. Nevertheless, it was felt that a substantially revised paper could make contributions to the field.

Important points to address:

The following points raised by the reviewers must be addressed in the revised manuscript. Note that no additional experimentation is specifically asked for. These queries concern justification of statements, or clarification of interpretations. In some cases, the points made refer to issues that the reviewers feel are not well justified, and if the justification cannot be improved you may wish to remove the pertinent discussion or downplay it. Please provide a point-by-point response to these queries along with the revised manuscript.

1) This manuscript describes structural studies of the *Pyrococcus furiosus* archaellum machinery. Using a combination of electron cryo-tomography, sub-tomogram averaging, single-particle cryoEM and helical reconstruction, the authors generated a model of the machinery. This work complements similar studies of related systems published over the last couple of years, including:

Thermus thermophilus type IVa pilus system (*eLife*. 2015 May 21;4. doi: 10.7554/*eLife*.07380),

Myxococcus xanthus type IVa pilus system (Science. 2016 351(6278):aad2001. doi: 10.1126/science.aad2001),type IVb system of Vibrio cholerae (Nat Microbiol. 2017 2:16269. doi: 10.1038/nmicrobiol.2016.269),

Neisseria menigitidis type IVa pilus (Nat Commun. 2016 7:13015. doi: 10.1038/ncomms13015),

Ignicoccus hospitalis type IV pili (J Mol Biol. 2012;422(44):274-281, and Proc Natl Acad Sci U S A. 2016 113(31):10352-7. doi: 10.1073/pnas.1607756113),

Methanospirillum hungatei archaellum (Nat Microbiol. 2016 2:16222. doi: 10.1038/nmicrobiol.2016.222).

The conclusion that it provides the first model of the entire machinery is, however, overstated, since the authors themselves admit that some components cannot be identified. Much of the data is observational and assignments not confirmed by generation and imaging of isogenic mutants (e.g. FlaCDE) as has been done in other studies cited above. A "large portion of the motor… remains unassigned". We therefore suggest toning down the claim (both in the Abstract as well as in the main text) that a structural model of the entire archaellum is presented. A structural model of the entire complex would require having every piece of density assigned. The paper must be revised to address these limitations.

2) "Archaea are ubiquitous microorganisms that are thought to be the ancestors of eukaryotes (Zaremba-Niedzwiedzka et al., 2017)." This remains controversial. They should look at papers such as: Nasir, A., Kim, K.M., Da Cunha, V., and Caetano-Anollés, G. (2016). Arguments Reinforcing the Three-Domain View of Diversified Cellular Life. Archaea 2016. Further, Zaremba-Niedzwiedzka et al., 2017 has been called into question by a number of people, and at least one paper is in press providing alternate interpretations for the metagenomic data analyzed as coming from "Lokiarchaeota" (namely, that there is no single organism that was being sequenced).

3) (Introduction) "Proton motive force". In fact, some bacterial flagellar motors are powered by a Na gradient. They should replace "proton motive force" with "ion fluxes".

4) "As FlaJ is almost completely buried in the membrane (Reindl et al., 2013), it is concealed by the lipid bilayer in our average". This makes no sense. It is not as if the authors are using metal shadowing to look at these structures. They are doing cryo-ET, and it is impossible to understand how the bilayer can be "concealing" the protein. They then go on to dock an x-ray structure into the density that is apparently "concealed".

5) (Subsection “Location of motor subunits”) The model for the FlaI-FlaH complex is quite weak due to the limited resolution. Can a better justification for the model be provided?

6) (Subsection “Structure of the *P. furiosus* archaellum”) The model for the FlaB filament used the *M. hungatei* structure as a reference. Nowhere is there any comparison between their *P. furiosus* filament and the previous structures from *M. hunatei* or *I. hospitalis*. How similar are these? What is conserved among the three and what varies?

7) (Discussion) "We suggest that FlaF does not form a continuous channel." In membrane biology, "channel" has a very specific meaning, and it appears that the authors are not referring to a pore of any sort. Further, it is also possible that the arguments in Banerjee et al. 2015 are flawed. The main argument for this location of FlaF seems to be that it bound S-layer proteins in vitro. But an obvious control, of showing that FlaB does not bind these proteins, was never done.

8) (Subsection “Assembly of the archaellum machinery”, fourth paragraph) The suggestion that some/many chemoreceptors form hexagonal arrays, and therefore that the hexagonal arrays found in this study may be an as yet unknown chemoreceptor, appears pretty weak.

9) (Discussion) "…Classes the *P. furiosus* archaellum as T4P-like filament". The homology between T4P and archaeal flagellar filaments only involves the N-terminal α-helical domain, including the leader sequence and the processing of this leader by a specific protease. There is no apparent homology, from either sequence or structure, between the large globular domain in archaeal flagellar filaments and any T4P. The "versatility" of the C-terminal domain is due to a gene fusion!

10) (Discussion) "Bacteria on the other hand have developed the far more complex flagellum based on the type-3 secretion system instead (Chen et al., 2011; Erhardt, Namba and Hughes, 2010)." We do not know of anyone other than the authors who thinks that the flagellum evolved from a T3SS. In fact, if one looks at reference (Erhardt, Namba and Hughes, 2010) which is cited, it states that the dogma in the field has been that the T3SS evolved from flagellar, but some new work suggests a common ancestor for both.

11) (Subsection “Electron cryo-tomography of Pyrococcus furiosus”, last paragraph) The polar cap seems to be associated with the machinery but its composition is unknown. If it is only present when archaella are present, is it possible to speculate on its composition based on genetic conservation in other species that also have this structure? Are there unassigned ORFs in archaella operons that might encode its constituents?

12) (Subsection “Integration of the archaellum into the periplasm”, last paragraph) The idea that S-layer subunits would need to be missing to accommodate the filament's passage in the absence of a dedicated channel requires better support. How likely is it that a hyperthermophile could tolerate missing S-layer subunits – would this destabilize the array? What evidence is there that such gaps actually occur? Do other surface structures use native gaps such as these?

13) The argument that the angles at which filaments cross the periplasm supports random egress needs better support as well; what is the chance that angled filaments are an artefact of processing? How would filaments emerging at random angles affect archaellum function and coordination? Could swimming occur efficiently if the archaella are not aligned?

14) (Subsection “Location of motor subunits”, first paragraph) Calling FlaJ a "secretin" is confusing, as this term is associated with outer membrane oligomers in gram negative type IV pilus and type II secretion systems. Instead it appears to be orthologous to the platform protein PilC that coordinates subunit assembly in related systems.

15) (Subsection “Sub-cellular organisation of motor complexes and the polar cap”, last paragraph) Averaging of only 57 units seems low – what is the confidence level of this information?

16) (Subsection “Structure of the *P. furiosus* archaellum”, second paragraph) The assignment of FlaB0 as the main subunit should be supported by imaging of mutants, or stated with less certainty.

17) (Subsection “Structure of the *P. furiosus* archaellum”, last paragraph) The angle at which the helices are arranged should be stated more clearly here. How variable is this arrangement?

18) (Subsection “N-glycosylation of the *P. furiosus* archaellum”) What is the justification for the assignment of N-glycans to these densities? Where they confirmed by mutation of one or more sites to a non-acceptor? Although complete loss of glycosylation prevents assembly, it is possible to reduce the number of sequons without impeding filament formation.

19) (Discussion) The Discussion could better position the new information in context with what has been observed in other systems as listed above. How do the details described here advance the field? How do the systems compare? How representative of archaella machineries is this one? What does "coordinated by FlaF" mean? This is a vague term.

20) (Subsection “Assembly of the archaellum machinery”, fourth paragraph) The paper provides no data supporting the "docking station" concept, and chemoreceptors usually have surface exposed ligand binding domains, which I didn't see here as the polar cap is positioned well below the membrane.

21) (Subsection “Assembly of the archaellum machinery”, last paragraph) Can the functional properties conferred by glycosylation be deconvolved from its requirement for filament polymerization? If they don't make filaments in the absence of glycosylation, one cannot conclude that glycosylation is important for adherence.

22) (Subsection “Evolutionary aspects”, last paragraph) Type IV pili do not rotate – their motors do.

23) Is there a hole visible in the docked FlaI structures (Figure 2, gold) and what is the diameter? Could the entire filament (diameter 11nm) fit into the FlaI ring or maybe only the helical core?

24) The last figure (Figure 5—figure supplement 4; transmembrane helix prediction) is not necessary and should be deleted.

25) For the filament structure at 4.2 A, a typical "Table 1" should be provided with information on data collection statistics, refinement statistics and model quality descriptors (Geometry, Ramachandran, Clash, Molprobity), ideally also EMRinger score, etc. It should also be described what kind of additional restraints were used in the Phenix refinement (e.g. Ramachandran, secondary structure, etc.).

26) "It is conceivable that the polar cap functions to concentrate archaella at one cell pole and acts as an anchor to prevent mechanical disruption of the cytoplasmic membrane by the rapid rotation of multiple archaella." One might assume that the forces needed to rotate the filaments are very large compared to the forces needed to rotate the (unfixated) motor complex in the lipid bilayer. Rather than just prevent disruption, one would think that the polar cap is required first of all to fix the motors, otherwise only the motors would rotate and the long filaments would stand still; in particular since there does not seem to be any anchoring to the periplasm. Is there any idea why there is a kink in the polar cap?

---

## [Author Response]

Important points to address:

*The following points raised by the reviewers must be addressed in the revised manuscript. Note that no additional experimentation is specifically asked for. These queries concern justification of statements, or clarification of interpretations. In some cases, the points made refer to issues that the reviewers feel are not well justified, and if the justification cannot be improved you may wish to remove the pertinent discussion or downplay it. Please provide a point-by-point response to these queries along with the revised manuscript.*

*1) This manuscript describes structural studies of the Pyrococcus furiosus archaellum machinery. Using a combination of electron cryo-tomography, sub-tomogram averaging, single-particle cryoEM and helical reconstruction, the authors generated a model of the machinery. This work complements similar studies of related systems published over the last couple of years, including:*

*Thermus thermophilus type IVa pilus system (eLife. 2015 May 21;4. doi: 10.7554/eLife.07380),*

*Myxococcus xanthus type IVa pilus system (Science. 2016 351(6278):aad2001. doi: 10.1126/science.aad2001),type IVb system of Vibrio cholerae (Nat Microbiol. 2017 2:16269. doi: 10.1038/nmicrobiol.2016.269),*

*Neisseria menigitidis type IVa pilus (Nat Commun. 2016 7:13015. doi: 10.1038/ncomms13015),*

*Ignicoccus hospitalis type IV pili (J Mol Biol. 2012;422(44):274-281, and Proc Natl Acad Sci U S A. 2016 113(31):10352-7. doi: 10.1073/pnas.1607756113),*

*Methanospirillum hungatei archaellum (Nat Microbiol. 2016 2:16222. doi: 10.1038/nmicrobiol.2016.222).*

*The conclusion that it provides the first model of the entire machinery is, however, overstated, since the authors themselves admit that some components cannot be identified. Much of the data is observational and assignments not confirmed by generation and imaging of isogenic mutants (e.g. FlaCDE) as has been done in other studies cited above. A "large portion of the motor… remains unassigned". We therefore suggest toning down the claim (both in the Abstract as well as in the main text) that a structural model of the entire archaellum is presented. A structural model of the entire complex would require having every piece of density assigned. The paper must be revised to address these limitations.*

We have toned down statements that suggest that we have generated a model of the entire archaellum complex by modifying the following sentences:

“This allows us to build a structural model of the entire archaellum, paving the way to a molecular understanding of archaeal swimming motion.”

Changed to: “This allows us to build a structural model combining the archaellum and its motor complex, paving the way to a molecular understanding of archaeal swimming motion.”

“In combination, our results provide a first structural model of the entire archaellar locomotion machinery.”

Changed to: “In combination, our results provide a first model of the architecture of the archaellar locomotion machinery.”

“Together with a wealth of previous biochemical data, our new structures of the archaellar motor complex, the polar cap, the S-layer, and the archaellum itself enable us to build a first structural model of the entire archaellum machinery (Figure 7).”

Changed to: “Together with a wealth of previous biochemical data, our new structures of the archaellar motor complex, the polar cap, the S-layer, and the archaellum itself enable us to build a first model describing the architecture of the archaellum machinery (Figure 7).”

*2) "Archaea are ubiquitous microorganisms that are thought to be the ancestors of eukaryotes (Zaremba-Niedzwiedzka et al., 2017)." This remains controversial. They should look at papers such as: Nasir, A., Kim, K.M., Da Cunha, V., and Caetano-Anollés, G. (2016). Arguments Reinforcing the Three-Domain View of Diversified Cellular Life. Archaea 2016. Further, Zaremba-Niedzwiedzka et al., 2017 has been called into question by a number of people, and at least one paper is in press providing alternate interpretations for the metagenomic data analyzed as coming from "Lokiarchaeota" (namely, that there is no single organism that was being sequenced).*

We have deleted the statement to avoid controversy.

*3) (Introduction) "Proton motive force". In fact, some bacterial flagellar motors are powered by a Na gradient. They should replace "proton motive force" with "ion fluxes".*

We have changed the wording as suggested.

*4) "As FlaJ is almost completely buried in the membrane (Reindl et al., 2013), it is concealed by the lipid bilayer in our average". This makes no sense. It is not as if the authors are using metal shadowing to look at these structures. They are doing cryo-ET, and it is impossible to understand how the bilayer can be "concealing" the protein. They then go on to dock an x-ray structure into the density that is apparently "concealed".*

There are many examples where sub-tomogram averaging of cryo-tomograms did not clearly reveal membrane-buried protein components, especially when native biological systems such as cells or isolated membranes were imaged. This limitation is mainly a consequence the high apparent contrast of lipid bilayers in tomograms at high defocus. To avoid confusion, we changed the wording as follows:

“As FlaJ is almost completely buried in the membrane, it is concealed by the lipid bilayer in our average (Figure 2).”

Changed to: “FlaJ is almost completely buried in the lipid bilayer. Due to the high apparent contrast of lipid bilayers in tomograms at high defocus, it was invisible in our average (Figure 2).”

In addition, we would like to clarify that we did not dock structures into the membrane. Instead, we docked the X-ray structures of FlaI and FlaH into the cytosolic part of the complex, which is well resolved. To avoid confusion, we have added the phrase “cytosolic part”:

“However, a superposition of the X-ray structure of the highly conserved *S. acidocaldarius* FlaI (SaFlaI; PDB-4IHQ) with the membrane-proximal cytosolic part of the sub-tomogram average places the X-ray structure into a density that closely matches its outline.”

*5) (Subsection “Location of motor subunits”) The model for the FlaI-FlaH complex is quite weak due to the limited resolution. Can a better justification for the model be provided?*

The orientation of FlaI is based on the interaction known from type II secretion ATPases with their cognate membrane protein. The orientation of the FlaH hexamer was guided by experiments that have not been published yet, but which indicate that this is the interaction interface. We intend to publish these results in a follow-up study.

*6) (Subsection “Structure of the P. furiosus archaellum”) The model for the FlaB filament used the M. hungatei structure as a reference. Nowhere is there any comparison between their P. furiosus filament and the previous structures from M. hunatei or I. hospitalis. How similar are these? What is conserved among the three and what varies?*

We have created two new supplementary figures (Figure 5—figure supplement 5 and Figure 5—figure supplement 6) and added two new paragraphs to the Results (–subsection “Structure of the *P. furiosus* archaellum”, last paragraph) as well as to the Discussion (subsection “Assembly of the archaellum machinery”, fifth paragraph), in which we now compare these three filaments and their component proteins in detail. We would like to emphasise that the filaments of *P. furiosus* and *M. hungatei* are both true archaella and more closely related to each other than to the Iho670 fibre of the immotile *I. hospitalis*, which is most likely an adhesive filament.

*7) (Discussion) "We suggest that FlaF does not form a continuous channel." In membrane biology, "channel" has a very specific meaning, and it appears that the authors are not referring to a pore of any sort. Further, it is also possible that the arguments in Banerjee et al. 2015 are flawed. The main argument for this location of FlaF seems to be that it bound S-layer proteins in vitro. But an obvious control, of showing that FlaB does not bind these proteins, was never done.*

The function of FlaF is not known, but it is the only archaellum protein which has been shown to interact with an extracellular component of the cell, the S-layer (Banerjee et al., Structure, 2015) and it was therefore proposed that it helps to anchor the archaellum in the cell envelope. We are not sure how to interpret the comment of the reviewer that these results might be “flawed”. As a control, the cytoplasmic FlaI was shown not to interact with the S-layer and FlaB was not part of the mentioned study. However, it is indeed not known how FlaF interacts with the S-layer or FlaG or FlaB. All these three proteins exhibit a globular domain which is predicted as archaellin domain, and the FlaF structure was even used to model the *Ignicoccus* Type IV pilus filament. Therefore, it is very likely, that all three proteins, FlaG/F and FlaB are extracellular as proven for FlaF and FlaB.

To avoid confusion with (ion) channels, we changed the in several places of the manuscript:

“…[H]owever it is not certain that this protein forms a bona fide channel-like complex through the periplasm.”

Changed to: “…however it is not certain that this protein forms a bona fide conduit through the periplasm.”

“Since S-layers are two-dimensional porous protein lattices, we wondered if the S-layer pores themselves form channels for archaellar filaments.”

Changed to: “Since S-layers are two-dimensional porous protein lattices, we wondered if the S-layer pores themselves can form conduits for archaellar filaments.”

“…[W]e suggest that FlaF does not form a continuous channel…”

Changed to: “…we suggest that FlaF does not form a continuous conduit that connects the membrane and the outer canopy of the S-layer.”

“Whereas the archaellar motor lacks a complex channel through the periplasm…”

Changed to: “Whereas the archaellar motor apparently lacks a conduit through the periplasm…”

*8) (Subsection “Assembly of the archaellum machinery”, fourth paragraph) The suggestion that some/many chemoreceptors form hexagonal arrays, and therefore that the hexagonal arrays found in this study may be an as yet unknown chemoreceptor, appears pretty weak.*

We have toned down the discussion concerning the identity of the hexagonal protein arrays as follows:

“The protein arrays found in this study may thus represent a yet unknown *Pyrococcus*-specific type of chemoreceptor.”

Changed to: “It remains to be investigated if the hexagonal protein arrays found in this study represent an unknown *Pyrococcus*-specific type of chemoreceptor or some other protein linked to motility.”

*9) (Discussion) "…Classes the P. furiosus archaellum as T4P-like filament". The homology between T4P and archaeal flagellar filaments only involves the N-terminal α-helical domain, including the leader sequence and the processing of this leader by a specific protease. There is no apparent homology, from either sequence or structure, between the large globular domain in archaeal flagellar filaments and any T4P. The "versatility" of the C-terminal domain is due to a gene fusion!*

We agree with the reviewer have re-written the Discussion accordingly (“Evolutionary aspects Evolutionary aspects”).

*10) (Discussion) "Bacteria on the other hand have developed the far more complex flagellum based on the type-3 secretion system instead (Chen et al., 2011; Erhardt, Namba and Hughes, 2010)." We do not know of anyone other than the authors who thinks that the flagellum evolved from a T3SS. In fact, if one looks at reference (Erhardt, Namba and Hughes, 2010) which is cited, it states that the dogma in the field has been that the T3SS evolved from flagellar, but some new work suggests a common ancestor for both.*

We have rephrased our statement accordingly:

“Bacteria on the other hand have developed the far more complex flagellum based on the type-3 secretion system instead”

Changed to: “For swimming motion, Bacteria have developed the flagellar machinery, a massive double-membrane spanning macromolecular device that is thought to share a common ancestor with the bacterial type-3 secretion system”.

*11) (Subsection “Electron cryo-tomography of Pyrococcus furiosus”, last paragraph) The polar cap seems to be associated with the machinery but its composition is unknown. If it is only present when archaella are present, is it possible to speculate on its composition based on genetic conservation in other species that also have this structure? Are there unassigned ORFs in archaella operons that might encode its constituents?*

The *Pyrococcus furiosus* archaellum (fla-) operon does not show unassigned regions that could encode for the polar cap. We suspect that this structure may be found in regions neighbouring the *Pfu* fla-operon that encode for putative proteins or may be under control of the fla-operon transcriptional regulator (EarA; Ding et al., Mol Microbiol, 2016). As blast search of these genes did not reveal significant homology with other known proteins, we are currently attempting to isolate the polar cap structure to analyse its composition via mass spectrometry.

*12) (Subsection “Integration of the archaellum into the periplasm”, last paragraph) The idea that S-layer subunits would need to be missing to accommodate the filament's passage in the absence of a dedicated channel requires better support. How likely is it that a hyperthermophile could tolerate missing S-layer subunits – would this destabilize the array? What evidence is there that such gaps actually occur? Do other surface structures use native gaps such as these?*

S-layers are inherently imperfect and highly porous protein lattices. They contain frequent symmetry breaks and gaps, which in fact are prerequisites for S-layers to form closed, cage-like structures around archaeal cells that are often irregularly shaped. In addition, gas in S-layers are also required for cell division (see Pum et al., J. Bacteriol., 1991). Missing subunits are therefore very common. We have included the following statement:

“Symmetry breaks and gaps are common in S-layers, as they enable them to form closed cages around cells and are essential for cell division. However, the mechanism by which the S-layer lattice is locally dissolved to provide sufficient space for the growing archaellum filament is still unknown and awaits further investigation.”

*13) The argument that the angles at which filaments cross the periplasm supports random egress needs better support as well; what is the chance that angled filaments are an artefact of processing? How would filaments emerging at random angles affect archaellum function and coordination? Could swimming occur efficiently if the archaella are not aligned?*

Although archaella appear to traverse the periplasm at random angles, they are in fact perpendicular to the plane of the polar cap (Briegel et al., bioRxiv, 2017). We have made the same observations in *Methanocaldococcus villosus* with cells that were high-pressure frozen or chemically fixed before embedding and sectioning (unpublished results) and in frozen-hydrated *P. furiosus* cells (this paper). Thus, it is unlikely that the angular variability of the archaella is an artefact of sample preparation.

This angular variability of the *P. furiosus* archaella is also fully consistent with both, the missing periplasmic conduit as well as the absence of a structure analogous to the bacterial flagellar hook. In contrast to bacterial flagella, archaella are slightly bent along their entire length and are not guided through the S-layer by a specific complex. Thus, it is not surprising that a slight bending in the periplasm is observed. In the absence of a hook, the inherent curvature is likely to increase the efficiency of rotary propulsion. Indeed, *P. furiosus* is a very efficient swimmer (Herzog & Wirth, Appl. Environ. Biol., 2012). However, as all these points are speculative, we would prefer not to elaborate on them in the manuscript.

*14) (Subsection “Location of motor subunits”, first paragraph) Calling FlaJ a "secretin" is confusing, as this term is associated with outer membrane oligomers in gram negative type IV pilus and type II secretion systems. Instead it appears to be orthologous to the platform protein PilC that coordinates subunit assembly in related systems.*

We have rephrased the wording from “secretin” to “platform protein”.

*15) (Subsection “Sub-cellular organisation of motor complexes and the polar cap”, last paragraph) Averaging of only 57 units seems low – what is the confidence level of this information?*

The arrays have six-fold symmetry, and we applied this symmetry during sub-tomogram averaging. Thus, the final number of averaged sub-volumes was 342, in line with other sub-tomogram averaging studies and yields a significant level of confidence.

*16) (Subsection “Structure of the P. furiosus archaellum”, second paragraph) The assignment of FlaB0 as the main subunit should be supported by imaging of mutants, or stated with less certainty.*

FlaB_0_ has already been shown to be the main constituent of the *P. furiosus* archaellum in a previous study involving using N-terminal sequencing and RT-PCR (Näther-Schindler et al., Front Microbiol, 2014).

Nevertheless, we toned down the corresponding statement in the manuscript:

“This shows that at least the main part of the filament is composed of FlaB_0_ rather than a mixture of all three FlaB proteins, confirming biochemical data that identified FlaB_0_ as major archaellin of *P. furiosus*”.

Changed to: “This suggests that at least the main part of the filament is composed of FlaB_0_ rather than a combination of all three FlaB proteins, in accordance with previous biochemical data that identified FlaB_0_ as the major archaellin of *P. furiosus*”

*17) (Subsection “Structure of the P. furiosus archaellum”, last paragraph) The angle at which the helices are arranged should be stated more clearly here. How variable is this arrangement?*

The helices are arranged at an angle of 15 degrees with respect to the axis of the filament. This has now been stated in the fourth paragraph of the subsection “Structure of the *P. furiosus* archaellum”. Judging from our structure, this arrangement is not variable.

*18) (Subsection “N-glycosylation of the P. furiosus archaellum”) What is the justification for the assignment of N-glycans to these densities? Where they confirmed by mutation of one or more sites to a non-acceptor? Although complete loss of glycosylation prevents assembly, it is possible to reduce the number of sequons without impeding filament formation.*

N-glycosylation is the most common form of glycosylation in Archaea and it has been shown previously that most archaella are highly glycosylated (Nikhil et al., FEMS Microbiol Rev, 2001). It has also been demonstrated by Periodic acid-Schiff staining that *P. furiosus* archaella are indeed glycosylated (Näther et al., J Bac, 2006). In our structure of the archaellum, all asparagine residues of FlaB_0_ that are part of a N-X-S/T (N-glycosylation) sequon, showed a large density next to these residues that could not be assigned by the polypeptide backbone or another side chain. Similar densities have been attributed to glycosylation sites of the archaellum from *M. hungatei* before (Poveleit et al., Nat. Microbiol., 2016). Conversely, none of the asparagines outside a N-X-S/T sequon showed such a large neighbouring densities in our structure. Thus, we are confident about our conclusion that these sites are indeed N-glycosylated. Generating new high-resolution structures of site-specific mutants is obviously not within the scope of this study but will be subject of future research.

*19) (Discussion) The Discussion could better position the new information in context with what has been observed in other systems as listed above. How do the details described here advance the field? How do the systems compare? How representative of archaella machineries is this one? What does "coordinated by FlaF" mean? This is a vague term.*

We have included a comparison between the bacterial T4P machinery and the archaellar motor in the Discussion, subsection “Evolutionary aspects”, second paragraph.

We have also included two new figures to compare the *P. furiosus* archaellum with that of *M. hungatei* and the *I. hospitalis* fibre (Figure 5—figure supplement 5 and Figure 5—figure supplement 6) and added two new paragraphs comparing these three filaments in the Results (subsection “Structure of the P. furiosus archaellum”, last paragraph) as well as the Discussion section (subsection “Structure of the *P. furiosus* archaellum”).

We have also included a statement highlighting the significance of our model for the field in the last paragraph of the subsection “Assembly of the archaellum machinery”.

*20) (Subsection “Assembly of the archaellum machinery”, fourth paragraph) The paper provides no data supporting the "docking station" concept, and chemoreceptors usually have surface exposed ligand binding domains, which I didn't see here as the polar cap is positioned well below the membrane.*

We have changed the wording to avoid the speculative term “docking station”.

“In addition, the polar cap appears to be a docking station for other protein complexes that form hexameric protein arrays.”

Changed to: “In addition, the polar cap is associated with protein complexes that form hexameric protein arrays.”

We have also toned down the discussion concerning the identity of the hexagonal protein arrays as follows:

“The protein arrays found in this study may thus represent a yet unknown Pyrococcus-specific type of chemoreceptor.”

Changed To: “It remains to be investigated if the hexagonal protein arrays found in this study represent an unknown Pyrococcus-specific type of chemoreceptor or some other protein linked to motility.”

*21) (Subsection “Assembly of the archaellum machinery”, last paragraph) Can the functional properties conferred by glycosylation be deconvolved from its requirement for filament polymerization? If they don't make filaments in the absence of glycosylation, one cannot conclude that glycosylation is important for adherence.*

We have separated the role of glycans in filament stability and adhesion by rephrasing the corresponding paragraph as follows:

“The extensive glycan cover of the archaellum filaments likely increases their stability. […] Finally, surface glycans also provide strain-specific recognition signatures for cell-cell interactions, in which archaella seem to play a major role.

*22) (Subsection “Evolutionary aspects”, last paragraph) Type IV pili do not rotate – their motors do.*

In fact, the cited paper (Nivaskumar, Structure, 2014) does show that pseudopili are assembled in a “spool-like” fashion and thus rotate during assembly. We have amended our wording for clarity:

“Although primordial T4P were likely not used for rotary propulsion, it has been proposed that pseudopili of T2SS, an evolutionary relative of the archaellar motor, rotate during assembly”.

Changed to: “Although it is unknown if these primordial filaments were used for rotary propulsion, it has been proposed that pseudopili of T2SS, an evolutionary relative of the archaellar motor, assemble in a rotary “spool-like” fashion.”

*23) Is there a hole visible in the docked FlaI structures (Figure 2, gold) and what is the diameter? Could the entire filament (diameter 11nm) fit into the FlaI ring or maybe only the helical core?*

There is indeed a hole in the docked FlaI X-ray structure, which in theory would be large enough to accommodate the helical core of the filament. However, it is unlikely that the filament protrudes into this hole as the helices at its end are fully embedded in the lipid bilayer, while FlaI is positioned in the cytoplasm. It is more likely that the helical core of the filament interacts directly with the membrane-integral platform protein FlaJ during assembly and rotation.

*24) The last figure (Figure 5—figure supplement 4; transmembrane helix prediction) is not necessary and should be deleted.*

We do think that this figure is important, as we refer to it in the Results section and have therefore decided to leave it in the manuscript.

*25) For the filament structure at 4.2 A, a typical "Table 1" should be provided with information on data collection statistics, refinement statistics and model quality descriptors (Geometry, Ramachandran, Clash, Molprobity), ideally also EMRinger score, etc. It should also be described what kind of additional restraints were used in the Phenix refinement (e.g. Ramachandran, secondary structure, etc.).*

Table 1 has been included and is referenced in the Results and Materials and methods sections.

*26) "It is conceivable that the polar cap functions to concentrate archaella at one cell pole and acts as an anchor to prevent mechanical disruption of the cytoplasmic membrane by the rapid rotation of multiple archaella." One might assume that the forces needed to rotate the filaments are very large compared to the forces needed to rotate the (unfixated) motor complex in the lipid bilayer. Rather than just prevent disruption, one would think that the polar cap is required first of all to fix the motors, otherwise only the motors would rotate and the long filaments would stand still; in particular since there does not seem to be any anchoring to the periplasm. Is there any idea why there is a kink in the polar cap?*

We have incorporated this reviewer’s suggestion into our Discussion.

“It is conceivable that the polar cap functions in concentrating archaella at one cell pole and acts as an anchor to fix the motor complexes in the bilayer to prevent futile rotation. In addition, the polar cap may also prevent the mechanical disruption of the cytoplasmic membrane by the rapid rotation of multiple archaella.”